# BRAF$^{V600E}$ cooperates with CDX2 inactivation to promote serrated colorectal tumorigenesis

Naoya Sakamoto[1,2], Ying Feng[1], Carmine Stolfi[1], Yuki Kurosu[1], Maranne Green[1], Jeffry Lin[1], Megan E Green[1], Kazuhiro Sentani[2], Wataru Yasui[2], Martin McMahon[3,4], Karin M Hardiman[5], Jason R Spence[1,6], Nobukatsu Horita[7], Joel K Greenson[8], Rork Kuick[9], Kathleen R Cho[1,8], Eric R Fearon[1,8,10]*

[1]Department of Internal Medicine, University of Michigan, Ann Arbor, United States; [2]Department of Molecular Pathology, Institute of Biomedical and Health Sciences, Hiroshima University, Hiroshima, Japan; [3]Department of Dermatology, University of Utah Medical School, Salt Lake City, United States; [4]Huntsman Cancer Institute, University of Utah Medical School, Salt Lake City, United States; [5]Department of Surgery, University of Michigan, Ann Arbor, United States; [6]Department of Cell and Developmental Biology, University of Michigan, Ann Arbor, United States; [7]Department of Molecular and Integrative Physiology, University of Michigan, Ann Arbor, United States; [8]Department of Pathology, University of Michigan, Ann Arbor, United States; [9]Department of Biostatistics, University of Michigan, Ann Arbor, United States; [10]Department of Human Genetics, University of Michigan, Ann Arbor, United States

*For correspondence: fearon@umich.edu

**Abstract** While 20–30% of colorectal cancers (CRCs) may arise from precursors with serrated glands, only 8–10% of CRCs manifest serrated morphology at diagnosis. Markers for distinguishing CRCs arising from 'serrated' versus 'conventional adenoma' precursors are lacking. We studied 36 human serrated CRCs and found CDX2 loss or *BRAF* mutations in ~60% of cases and often together (p=0.04). CDX2$^{Null}$/BRAF$^{V600E}$ expression in adult mouse intestinal epithelium led to serrated morphology tumors (including carcinomas) and BRAF$^{V600E}$ potently interacted with CDX2 silencing to alter gene expression. Like human serrated lesions, CDX2$^{Null}$/BRAF$^{V600E}$-mutant epithelium expressed gastric markers. Organoids from CDX2$^{Null}$/BRAF$^{V600E}$–mutant colon epithelium showed serrated features, and partially recapitulated the gene expression pattern in mouse colon tissues. We present a novel mouse tumor model based on signature defects seen in many human serrated CRCs – CDX2 loss and BRAF$^{V600E}$. The mouse intestinal tumors show significant phenotypic similarities to human serrated CRCs and inform about serrated CRC pathogenesis.

## Introduction

Colorectal cancers (CRC) have accumulated defects in oncogenes and tumor suppressor genes (TSGs) (*Fearon and Vogelstein, 1990*). Many CRCs likely arise via clonal outgrowth and evolution of an epithelial cell or cells in an adenomatous precursor lesion via so-called adenoma-carcinoma progression (*Fearon, 2011*; *Fearon and Vogelstein, 1990*). But, upwards of 20–30% of CRCs may arise via an alternative pathway, often termed the 'serrated pathway', to reflect the saw-tooth or serrated morphology of the epithelial glands in the presumptive benign precursor lesions

(*Bettington et al., 2013*; *Jass, 2007a*; *Langner, 2015*). Serrated benign lesions arising in the colon and rectum include hyperplastic polyps (HPPs), sessile serrated adenomas/polyps (SSAs/SSPs), and traditional serrated adenomas (TSAs) (*Bettington et al., 2013*; *Langner, 2015*). HPPs comprise the vast majority of serrated benign lesions detected and are most often found in the distal colon and rectum (*Bettington et al., 2013*; *Langner, 2015*). SSAs/SSPs account for 5–20% of serrated benign lesions and arise more often in the proximal colon (*Bettington et al., 2013*; *Langner, 2015*). TSAs account for only about 1% of serrated benign lesions and arise mostly in the distal colon and rectum (*Bettington et al., 2013*; *Langner, 2015*). Some SSAs/SSPs and TSAs, such as those showing dysplasia, have potential to progress to CRC (*Bettington et al., 2013*; *Jass, 2007a*; *Langner, 2015*). HPPs do not show dysplasia and have generally been assumed to have negligible malignant potential (*Bettington et al., 2013*; *Langner, 2015*). However, in rare patients who develop HPPs along with other serrated colon polyps or in other patients with multiple and large HPPs, the lifetime risk of CRC is elevated (*Jass, 2007b*). Hence, all three serrated benign lesion subtypes may have malignant potential, with SSAs and TSAs having higher potential than HPPs.

Recurrent somatic mutations in CRCs affect roughly two-dozen different genes (*Cancer Genome Atlas Network, 2012*). Mutations in the *APC* and *TP53* TSGs are found in about 80% and 60% of CRCs, respectively (*Cancer Genome Atlas Network, 2012*; *Fearon, 2011*). *KRAS* and *BRAF* oncogenic mutations are seen in roughly 40% and 10% of CRCs, respectively, and are mutually exclusive in a given CRC (*Cancer Genome Atlas Network, 2012*; *Fearon, 2011*). *KRAS* mutations most often result in missense substitutions at glycine codons 12 or 13, and *BRAF* mutations most often result in missense substitutions of glutamic acid (E) for valine (V) at codon 600 (V600E) (*Cancer Genome Atlas Network, 2012*; *Fearon, 2011*). Somatic *BRAF* mutations and loss or reduction of expression of the CDX2 homeobox transcription factor individually and concurrently have been associated with poor prognosis in CRC patients (*Bae et al., 2015*; *Clarke and Kopetz, 2015*; *Dalerba et al., 2016*; *Landau et al., 2014*).

*BRAF* mutations are found in 70–80% of SSAs/SSPs, whereas *KRAS* mutations are rare in these tumors (*Bettington et al., 2013*). *BRAF* mutations are found in about 70% of proximal colon HPPs and/or microvesicular histology HPPs, whereas *KRAS* mutations are found in up to 50% of HPPs in the distal colon and rectum and in goblet cell-rich histology HPPs. TSAs mostly have *KRAS* mutations (*Bettington et al., 2013*). In part because *BRAF* mutations are common in SSAs/SSPs and HPPs and not in conventional adenomas, it has been argued that CRCs arising from serrated precursor lesions may harbor *BRAF* mutations more often than CRCs arising from the conventional adenoma pathway (*Bettington et al., 2013*; *Langner, 2015*). *APC* mutations are rare or absent in serrated benign lesions and are perhaps also less common in CRCs arising from a serrated precursor (*Bettington et al., 2013*; *Langner, 2015*). Other molecular features suggested to be associated with serrated pathway CRCs include CpG island methylation phenotype high-frequency (CIMP-H) with $p16^{INK4A}$ and/or *MLH1* gene silencing, with *MLH1* silencing linked to the microsatellite instability-high (MSI-H) phenotype (*Bettington et al., 2013*; *Jass, 2007a*; *Langner, 2015*). Serrated pathway-type CRCs have also been suggested to express gastric epithelium markers, such as mucin 2 (MUC2), MUC5AC, MUC6 and annexin A10 (ANXA10) (*Kim et al., 2015*; *Tsai et al., 2015*; *Walsh et al., 2013*).

Although up to 20–30% of CRCs have been estimated to arise from benign serrated precursors (*Bettington et al., 2013*; *Jass, 2007a*; *Langner, 2015*), only 8–10% of CRCs manifest definitive serrated morphologic features at diagnosis, characterized by epithelial gland serration, intracellular and extracellular mucin, and absence of necrosis (*Bettington et al., 2013*; *García-Solano et al., 2010*; *Mäkinen, 2007*). CRCs with serrated morphology (hereafter referred to as 'serrated CRCs') are presumed to arise mostly from serrated precursor lesions (*Bettington et al., 2013*). Definitive biomarkers are lacking to identify the CRCs that arose from serrated precursors but lack serrated morphology at diagnosis. Importantly, clinicopathologic studies have suggested that serrated CRCs may pursue a more aggressive course than their conventional counterparts (*García-Solano et al., 2011*) Hence, to further define how molecular lesions may underlie the pathogenesis of CRCs arising from serrated precursors, we pursued in-depth molecular characterization of 36 CRCs with definite serrated morphological features at diagnosis, studying BRAF, KRAS, TP53, $p16^{INK4A}$, Wnt, and CDX2 pathway defects by DNA sequencing or protein expression and CIMP-H and MSI-H status by standard marker analyses. Because frequent concurrent loss of CDX2 expression and *BRAF* mutation suggested their cooperation in the pathogenesis of serrated CRCs, we modeled concurrent biallelic

inactivation of *Cdx2* (*Cdx2^{-/-}*) and expression of mutant BRAF$^{V600E}$ in adult mouse colon epithelium via conditional somatic gene targeting approaches. Tumors arising in the mice and organoids derived from gene-targeted epithelium were studied and compared to human serrated CRCs in order to inform knowledge of the molecular pathogenesis of serrated CRCs and to identify potential biomarkers of this group of poor prognosis tumors. Our novel mouse model should have utility for new prevention, diagnosis, and treatment approaches for this important biological and clinical group of CRCs.

## Results

### Loss of CDX2 expression and *BRAF* mutation are common and co-existing defects in human serrated morphology CRCs

We selected 36 CRCs meeting defined histopathological criteria for the diagnosis of serrated adeno-carcinoma (*Jass and Smith, 1992*; *Mäkinen et al., 2001*; *Tuppurainen et al., 2005*), with representative images of serrated morphology CRCs studied shown in *Figure 1—figure supplement 1*. We carried out analyses of selected molecular alterations in the 36 CRCs, focusing on the following: (i) DNA sequence-based determination of *KRAS* codons 12, 13, and 61 and *BRAF^{T1799A}*(encoding BRAF$^{V600E}$) missense mutations; (ii) DNA-based analyses of MSI-H and CIMP-H status; (iii) immuno-histochemical (IHC) studies of $\beta$-catenin, p53, MLH1, p16$^{INK4A}$, and CDX2 to inform about the expression and functional status of selected TSG pathways and proteins; and (iv) IHC studies of annexin A10 (ANXA10) and MUC5AC to inform about aberrant gastric epithelial marker expression. We found loss or marked reduction of CDX2 expression in 21 (58%) tumors (*Figure 1*). Representative staining patterns for CDX2 and other selected proteins and pathways under study are shown in *Figure 1—figure supplement 2*. Frequent alterations in the 36 CRCs included strong nuclear staining for p53 protein in neoplastic cells in 21 (58%) of cases, consistent with expression of a missense mutant p53 protein, and *BRAF^{T1799A}* missense mutation encoding BRAF$^{V600E}$ protein in 20 (56%) cases (*Figure 1*). Other changes seen included strong nuclear $\beta$-catenin staining, reflecting potential Wnt pathway mutations, in 19 (53%) cases; p16$^{INK4a}$ expression loss in 18 (50%) cases; the CIMP-H phenotype in 12 (33%) cases, with 8 of these 12 cases having lost *p16^{INK4A}* expression; the MSI-H phenotype in 5 (14%) cases; MLH1 expression loss in 4 (11%) cases, all of which were MSI-H cases; and *KRAS* codon 12 or 13 mutation in 7 (19%) cases, all of which lacked *BRAF^{T1799A}* mutations (*Figure 1*).

Because CDX2 loss/reduction was seen in 21 of the 36 (58%) CRCs and *BRAF* mutations were seen in 20 (56%) of cases, we sought to determine how CDX2 loss/reduction and *BRAF* mutations might be associated with specific clinical, pathological, or molecular features. We did not observe any statistically significant correlations between CDX2 loss/reduction or *BRAF^{T1799A}* mutation and patient age, tumor location, or selected tumor histological features, other than the absence of 'dirty necrosis' was associated with *BRAF* mutations (p=0.03) (*Figure 1—source data 1*). However, we did find that CDX2 loss/reduction and *BRAF* mutations were seen concurrently in 15 (42%) cases (p=0.04) (*Table 1* and *Figure 1*). Besides *BRAF* mutations, other features more commonly seen in CDX2-negative tumors were CIMP-H status (p=0.04); membranous staining for $\beta$-catenin throughout the tumor, including the invasive front, likely reflecting wild type status for Wnt pathway genes (e.g, *APC* and *CTNNB1*, encoding $\beta$-catenin) (p=0.05); and retention of p16$^{INK4A}$ expression (p=0.04) (*Table 1* and *Figure 1*). *BRAF* mutations in these cases were associated with membranous staining for $\beta$-catenin (p=0.003), CIMP-H status (p=0.004), and ANXA10 expression (p<0.0001) (*Table 1* and *Figure 1*). Interestingly, all 7 *KRAS*-mutant CRCs were CDX2-positive and lacked ANXA10 or MUC5AC expression, except for MUC5AC expression in one *KRAS*-mutant case (*Figure 1*).

### Combined *Cdx2* inactivation and BRAF$^{V600E}$ expression in mouse intestinal epithelium promotes serrated benign and invasive tumors

Because the human serrated morphology CRCs frequently showed marked reduction or loss of CDX2 expression and *BRAF^{T1799A}* mutations, we sought to address consequences of conditionally inactivating *Cdx2* or expressing the BRAF$^{V600E}$ oncoprotein kinase in mouse adult intestinal epithelium, as well as concurrent alterations of both *Cdx2* and BRAF$^{V600E}$. We previously described *CDX2P-CreER^{T2}* transgenic mice, expressing a tamoxifen (TAM)-regulated Cre protein (CreER$^{T2}$)

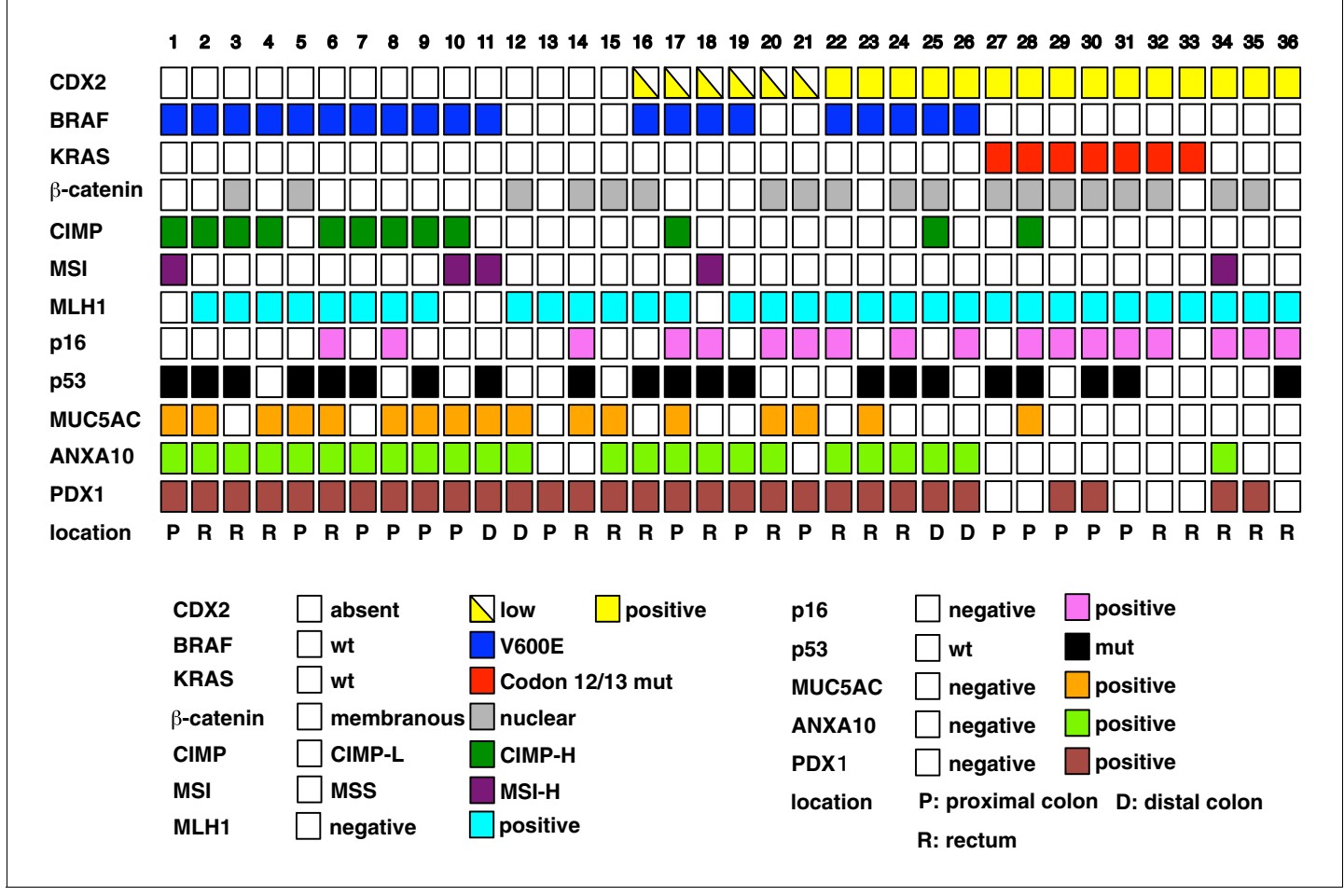

**Figure 1.** Overview of the result of immunohistochemistry (IHC) and somatic changes for selected proteins, genes, and phenotypic marker panels. Each square represents the results for IHCs and molecular phenotypic features in individual serrated adenocarcinoma cases.

The following source data and figure supplements are available for figure 1:

**Source data 1.** Clinicopathologic features, BRAF mutation status, and CDX2 expression in 36 human serrated morphology CRCs.

**Figure supplement 1.** Representative images of serrated adenocarcinoma cases.

**Figure supplement 2.** Representative pattern of immunohistochemistry in serrated adenocarcinoma (SAC) cases with CDX2-negative (upper) and CDX2-positive (lower) staining.

under control of human *CDX2* regulatory sequences (*Feng et al., 2013*). The transgene allows for TAM-inducible targeting of loxP-containing alleles in adult mouse terminal ileum, cecum, colon, and rectal epithelium. The median survival of *CDX2P-CreER^{T2} Cdx2^{fl/fl}* mice treated with TAM to activate Cre function and inactivate both *Cdx2^{fl}* alleles (*Blij et al., 2012*) in colon epithelium was greater than 480 days (*Figure 2A*). We also used the *CDX2P-CreER^{T2}* transgene to activate a conditionally oncogenic *Braf* allele (*Braf^{CA}*), which expresses normal BRAF until subjected to Cre-mediated recombination, resulting in expression of a mouse-human hybrid V600E oncogenic protein under the control of *Braf* endogenous regulatory elements (*Dankort et al., 2007*). Regardless of the species of origin, we refer to this mutationally activated form of BRAF as BRAF^{V600E} and the *Braf^{CA}* allele after Cre-mediated recombination as *Braf^{V600E}* for the sake of clarity. The median survival of *CDX2P-CreER^{T2} Braf^{CA}* mice was greater than 480 days after TAM treatment (*Figure 2A*). In marked contrast, the median survival of *CDX2P-CreER^{T2} Cdx2^{fl/fl} Braf^{CA}* mice was 103 days after TAM treatment to concurrently

**Table 1.** Immunohistochemistry and molecular features of 36 human serrated morphology CRCs.

| | CDX2 (+) | CDX2 (-) | *p value | BRAF mut | BRAF wt | *p value |
|---|---|---|---|---|---|---|
| Immunohistochemistry | | | | | | |
| MUC5AC(+) | 2 | 15 | 0.0008 | 11 | 6 | 0.335 |
| MUC5AC(-) | 13 | 6 | | 9 | 10 | |
| | | | | | | |
| MUC6(+) | 0 | 6 | 0.031 | 6 | 1 | 0.104 |
| MUC6(-) | 15 | 15 | | 14 | 15 | |
| | | | | | | |
| MUC2(+) | 9 | 17 | 0.260 | 13 | 13 | 0.456 |
| MUC2(-) | 6 | 4 | | 7 | 3 | |
| | | | | | | |
| MLH1 retained | 14 | 18 | 0.626 | 16 | 16 | 0.113 |
| MLH1 loss | 1 | 3 | | 4 | 0 | |
| | | | | | | |
| MSH2 retained | 15 | 21 | | 20 | 16 | |
| MSH2 loss | 0 | 0 | | 0 | 0 | |
| | | | | | | |
| $\beta$-catenin membranous | 4 | 13 | 0.049 | 14 | 3 | 0.003 |
| $\beta$-catenin nucleus | 11 | 8 | | 6 | 13 | |
| | | | | | | |
| p53 mut | 8 | 13 | 0.736 | 15 | 6 | 0.052 |
| p53 wt | 7 | 8 | | 5 | 10 | |
| | | | | | | |
| p16 retained | 11 | 7 | 0.041 | 7 | 11 | 0.092 |
| p16 loss | 4 | 14 | | 13 | 5 | |
| | | | | | | |
| Annexin A10 (+) | 6 | 18 | 0.010 | 20 | 4 | $1.4 \times 10^{-6}$ |
| Annexin A10 (-) | 9 | 3 | | 0 | 12 | |
| Molecular features | | | | | | |
| MSI | 1 | 4 | 0.376 | 4 | 1 | 0.355 |
| MSS | 14 | 17 | | 16 | 15 | |
| | | | | | | |
| CIMP phenotype (+) | 2 | 10 | 0.040 | 11 | 1 | 0.004 |
| CIMP phenotype (-) | 13 | 11 | | 9 | 15 | |
| Total | 15 | 21 | | 20 | 16 | |

*p values are from Fisher's exact test.

target the *Cdx2$^{fl/fl}$* and *Braf$^{CA}$* alleles (**Figure 2A**). The median survival of mice with simultaneous targeting of one *Apc$^{fl}$* allele together with *Cdx2$^{fl}$* and *Braf$^{CA}$* alleles was 158 days, indicating that inactivation of one *Apc* allele did not enhance tumor aggressiveness compared to the phenotype seen in *CDX2P-CreER$^{T2}$ Cdx2$^{fl/fl}$ Braf$^{CA}$* mice (**Figure 2A**).

In *CDX2P-CreER$^{T2}$ Braf$^{CA}$ mice* treated with TAM, we observed marked hyperplasia in more than 90% of crypts in the terminal ileum, cecum, and proximal colon (**Figure 2—figure supplement 1B**). The intestinal epithelium in *CDX2P-CreER$^{T2}$ Cdx$^{fl/fl}$* mice treated with TAM showed marked hyperplastic changes and irregular bifurcation of the crypts, but no dysplastic features (**Figure 2—figure supplement 1C**). When the TAM-treated *CDX2P-CreER$^{T2}$ Cdx2$^{fl/fl}$ Braf$^{CA}$ mice* were moribund, all of

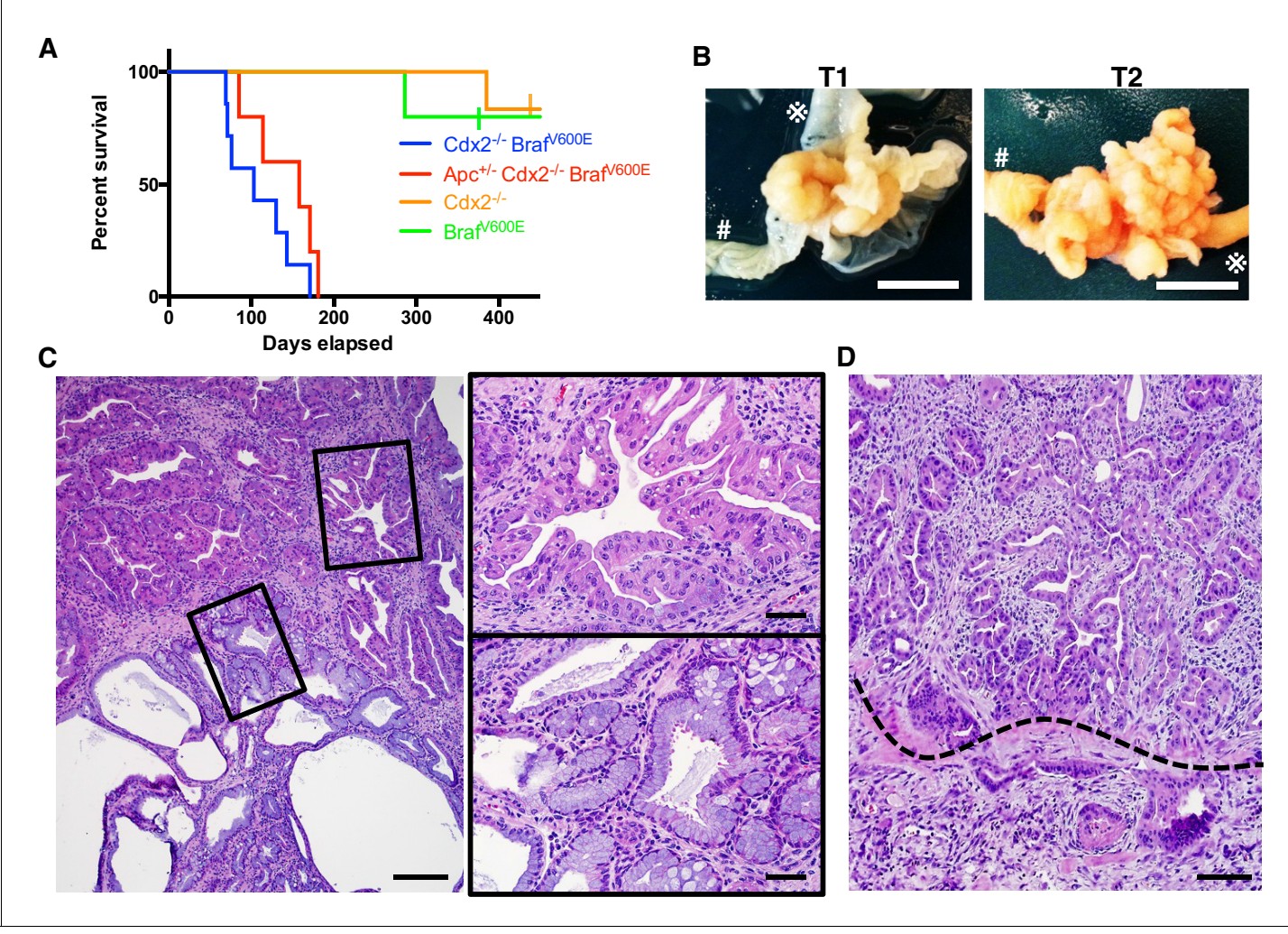

**Figure 2.** Characteristics of *CDX2P-CreER^T2 Cdx2^fl/fl Braf^CA* mice. (**A**) Kaplan-Meier curves of *CDX2P-CreER^T2 Cdx2^fl/fl Braf^CA* (*Cdx2^-/- Braf^V600E*, n = 7), *CDX2P-CreER^T2 Apc^fl/+Cdx2^fl/fl Braf^CA* (*Apc^+/- Cdx2^-/- Braf^V600E*, n = 5), *CDX2P-CreER^T2 Cdx2^fl/fl* (*Cdx2^-/-*, n = 6), and *CDX2P-CreER^T2 Braf^CA* (*Braf^V600E*, n = 5) mice, after two daily doses of TAM (150 mg/kg). p<0.002 when comparing *Cdx2^-/-* or *Braf^V600E* to *Cdx2^-/- Braf^V600E* or *Apc^+/- Cdx2^-/- Braf^V600E* by log-rank test; p=0.158 when comparing *Cdx2^-/- Braf^V600E* to *Apc^+/- Cdx2^-/- Braf^V600E* by log-rank test. (**B**) Macroscopic image of the tumors at the proximal colon-cecum-ileum junction of *CDX2P-CreER^T2 Cdx2^fl/fl Braf^CA* mice. Left: protuberant tumor (T1) located at the proximal colon-cecum-ileum junction; Right: polypoid lesions (T2) that occupied at the entire terminal ileum to cecum region. # indicates the proximal colon; ※ indicates the ileum. Scale bar, 10 mm (**C**) Histological features of the proximal colon-cecum-ileum junction tumor of *CDX2P-CreER^T2Cdx2^fl/fl Braf^CA* mice. Two major components were found in the tumor: serrated region (upper right) and mucin-rich region (lower left). Scale bars, 200 μm for low magnification image (left); 50 μm for high magnification images (right). (**D**) Serrated region showing invasion into submucosa. Dashed-line highlights muscular mucosa. Scale bar, 100 μm. See also *Figure 2—source data 1* for panel **A**.

The following source data and figure supplement are available for figure 2:

**Source data 1.** Raw data for Kaplan Meier analysis shown in *Figure 2A*.
**Source data 2.** Tumor size and invasion in *CDX2-CreER^T2 Cdx2^fl/fl Braf^CA* mice post TAM Injection.
**Figure supplement 1.** Combined *Cdx2* inactivation and *Braf^V600E* mutation in mouse intestinal epithelium promotes the formation of tumors with serrated morphology.

the mice had at least one large protuberant tumor (>10 mm) or multiple polypoid lesions at the proximal colon-ileal-cecal junction and each mouse had 2–6 independent tumor lesions with size >2 mm (*Figure 2B* and *Figure 2—source data 2*). All the tumors arising in *CDX2P-CreER^{T2} Cdx2^{fl/fl} Braf^{CA}* mice showed both serrated and mucin-rich histological components (*Figure 2C*). The serrated components had a distinct serrated architecture with irregular bifurcation of the glands, and the cells in these glands have eosinophilic and abundant cytoplasm, and vesicular nuclei with prominent nucleoli (*Figure 2C*). These characteristics closely resemble features of human serrated morphology CRCs. In the mucin-rich components, the glands were tubular or cystic, and lined by bland cuboidal and columnar cells containing abundant mucin (*Figure 2C*). Using DNA samples isolated from the serrated or mucin-rich regions by laser-capture microdissection, we found the mucin-rich components had only inactivated both *Cdx2^{fl}* alleles, whereas the serrated histology epithelium had both *Cdx2^{fl}* alleles and the *Braf^{CA}* allele targeted (*Figure 2—figure supplement 1D*). In the seven TAM-treated *CDX2P-CreER^{T2} Cdx2^{fl/fl} Braf^{CA}* mice studied, from 17–50% of the tumor lesions showed evidence that the serrated glands had invaded into the submucosa, indicative of carcinoma (*Figure 2D* and *Figure 2—source data 2*). Stochastic inactivation of the remaining wild type *Apc* allele in the setting of *Apc* hemizygous state in intestinal epithelium could potentially be seen as progression event in tumors initiated by other genetic drivers, such as in the setting of *Cdx2* biallelic inactivation and BRAF^{V600E} activation. We found that combined TAM-mediated targeting of an *Apc-fl* allele together with the *Cdx2^{fl}* and *Braf^{CA}* alleles in *CDX2P-CreER^{T2} Cdx2^{fl/fl} Braf^{CA} Apc^{fl/+}* mice did not alter the histology of the tumor lesions compared to those arising in *CDX2P-CreER^{T2} Cdx2^{fl/fl} Braf^{CA}* mice with intact *Apc* alleles (*Figure 2—figure supplement 1E*). Furthermore, we saw no apparent selection for loss of the remaining wild type *Apc* allele in tumors arising in *CDX2P-CreER^{T2} Cdx2^{fl/fl} Braf^{CA} Apc^{fl/+}* mice, as no β-catenin nuclear or cytoplasmic staining was seen in the *CDX2P-CreER^{T2} Cdx2^{fl/fl} Braf^{CA} Apc^{fl/+}* mouse tumor tissues studied (*Figure 2—figure supplement 1E*). Our findings indicate concurrent *Cdx2* silencing combined with BRAF^{V600E} expression in mouse intestinal epithelium leads to premature death of the mice, due to the development of intestinal tumors, including carcinomas, and the tumors recapitulate histological features of human serrated morphology CRCs.

## Molecular analyses of *Cdx2^{−/−}* and BRAF^{V600E}–expressing mouse serrated tumors

Akin to the situation in human serrated morphology CRCs with CDX2 loss and *BRAF^{T1799A}* mutations (*Figure 1*), ANXA10 expression was seen in the serrated components of the tumors arising in the TAM-treated *CDX2P-CreER^{T2} Cdx2^{fl/fl} Braf^{CA}* mice (*Figure 3A*). ANXA10 expression was not seen in any of the mucin-rich tumor components, where only *Cdx2* inactivation was present and BRAF^{V600E} expression was absent (*Figure 3A*). The BRAF protein functions as an upstream activator of mitogen-activated protein kinase (MAPK) signaling, and similar to ANXA10 expression, phospho-ERK expression, indicating MAPK activation, was detected only in the serrated components of tumors (*Figure 3A*). Similar to the infrequent nuclear or cytoplasmic β-catenin staining and strong MUC5AC expression seen in most CDX2-defective and BRAF^{V600E}-mutant human serrated CRCs (*Figure 1*), we found membranous β-catenin staining, strong MUC5AC expression, and bromo-deoxyuridine (BrdU) incorporation (indicating DNA synthesis) in the serrated components of the tumors in the *CDX2P-CreER^{T2} Cdx2^{fl/fl} Braf^{CA}* mice (*Figure 3B*). No evidence for p53 nuclear protein accumulation was seen in the serrated and/or invasive components (*Figure 3B*). Consistent with a previous study showing that EGFR activation was found in majority of CRC cell lines harboring *BRAF^{V600E}* mutation (*Prahallad et al., 2012*), the tumors arising in the TAM-treated *CDX2P-CreER^{T2} Cdx2^{fl/fl} Braf^{CA}* mice showed significant higher levels of phospho-EGFR expression, compared to very low level of phospho-EGFR in *Cdx2^{−/−}*-mutant, *Braf^{V600E}*-mutant, or normal control mouse cecal epithelium (*Figure 4*). All tumors from 5 *CDX2P-CreER^{T2} Cdx2^{fl/fl} Braf^{CA}* mice were MSS, and we did not detect significant methylation at the *p16^{Ink4a}* promoter in the tumors (*Figure 3—figure supplement 1*). Overall, the findings suggest concurrent CDX2 and BRAF^{V600E} defects are key instigating lesions in development of the serrated mouse tumors.

The canonical or β-catenin-dependent Wnt signaling pathway has a key role in regulating tissue stem cells near the small intestine and colon crypt base, in large part via regulation of various β-catenin/T cell factor (TCF) transcription pathway genes (*Koo and Clevers, 2014*). In most human adenomas and CRCs, Wnt pathway defects, most commonly *APC* inactivating mutations, lead to

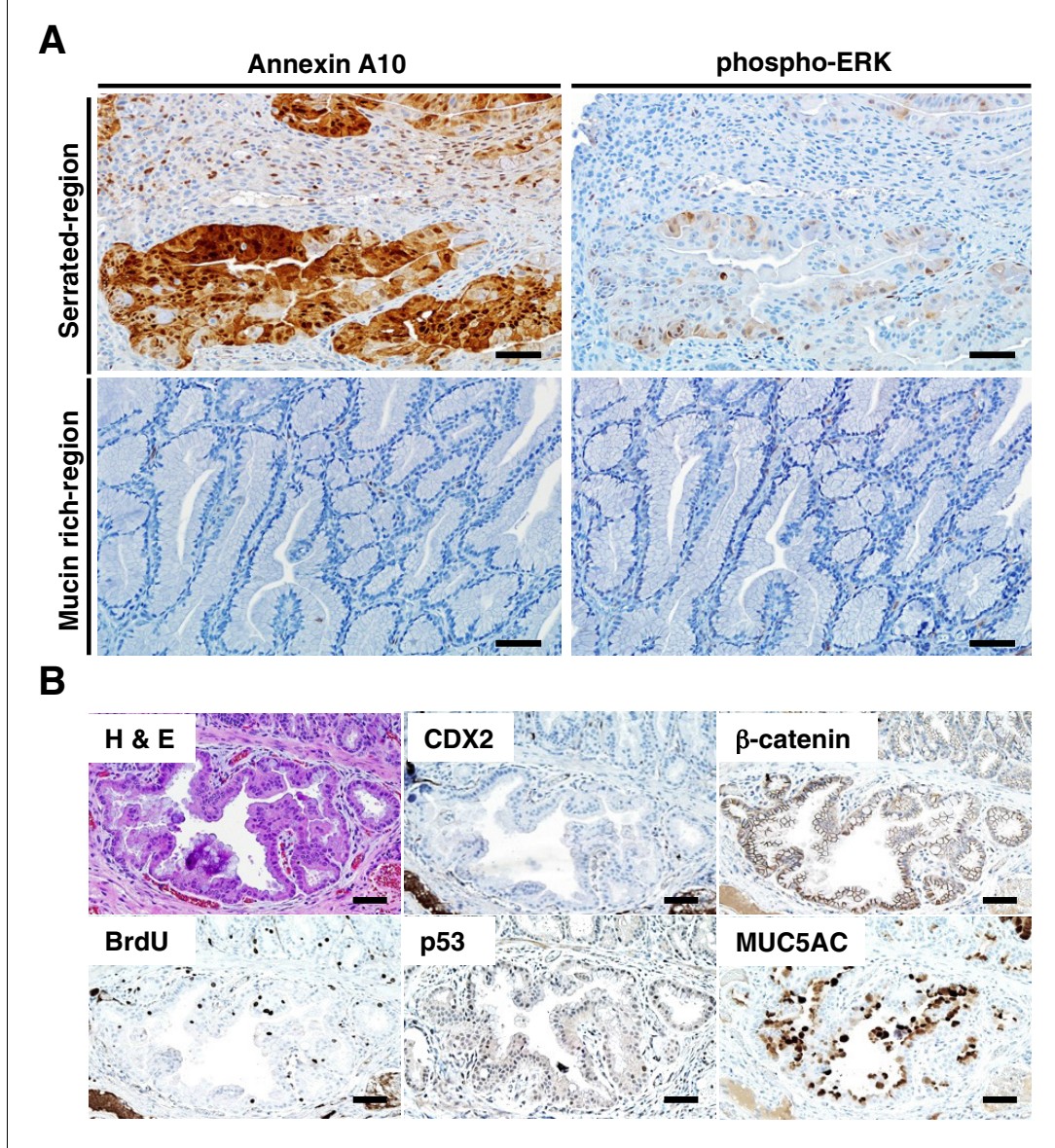

**Figure 3.** Immunohistochemical (IHC) staining in the tumors at the ileum-cecum junction of *CDX2P-CreER^{T2} Cdx2^{fl/fl} Braf^{CA}* mice. (**A**) Annexin A10 (left) and phospho-ERK (right) expression in serrated region (upper) and mucin-rich region (lower). Scale bars, 100 μm. (**B**) H and E (upper left) and IHC staining for CDX2 (upper middle), β-catenin (upper right), BrdU (lower left), p53 (lower middle) and MUC5AC (lower right) at the invasive region of the tumor. Scale bars, 50 μm.

The following source data and figure supplements are available for figure 3:

**Figure supplement 1.** Tumors from *CDX2P-CreER^{T2} Cdx2^{fl/fl} Braf^{CA}* mice were MSS, and did not show significant methylation at the *p16^{Ink4a}* promoter.

**Figure supplement 2.** Expression of intestinal stem cell markers, Wnt target genes and differentiation markers in mice.

**Figure supplement 2—source data 1.** Raw data for qRT-PCR analysis for mice shown in *Figure 3—figure supplement 2*.

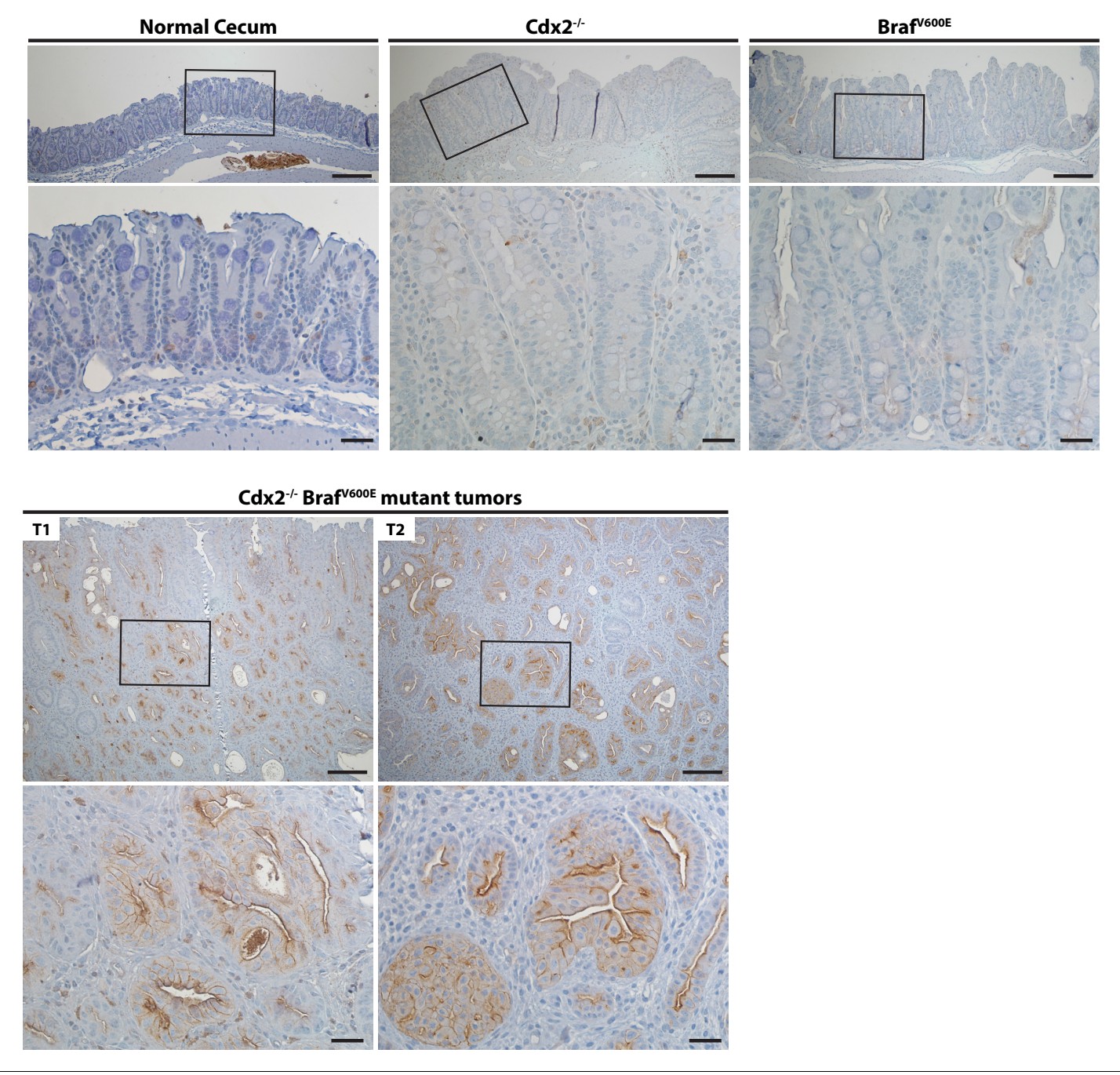

**Figure 4.** Activation of EGFR in tumor tissues of *CDX2-CreER^T2 Cdx2^fl/fl Braf^CA* mice. Two representative tumors from *CDX2-CreER^T2 Cdx2^fl/fl Braf^CA* mice (**T1** and **T2**) and cecum tissues from the TAM-treated Cre-negative control mouse (Normal cecum), the *CDX2P-CreER^T2* Cdx2^fl/fl (Cdx2^-/-) mouse, or the *CDX2P-CreER^T2 Braf^CA (Braf^V600E)* mouse were subjected to immunohistochemical (IHC) staining for active EGFR (phospho-EGFR Tyr845). Representative Imagines with low (top panels) and high (bottom panels) magnifications were shown. Scale bars, 100 µm for low magnification image (top); 20 µm for high magnification images (bottom).

dysregulated *β*-catenin/T cell factor (TCF) transcription with resultant activation of genes normally restricted to intestinal stem cells in many neoplastic cells in the lesions (*Fearon, 2011*; *Koo and Clevers, 2014*). We studied *Cdx2*-mutant, *Braf^V600E*-mutant, and *Cdx2 Braf^V600E* double-mutant, as well as *Apc*-mutant mouse colon epithelium, for expression of transcripts for various intestinal stem

cell markers and Wnt pathway genes, such as: *Olfm4, Lgr5, Msi1, Cd133, Lrig1, Hopx, Bmi1, Cd44, Axin2, Mmp7, Ephb2, cMyc* and *Sox9*. Compared to control colon tissues, only *Olfm4, Mmp7* and *Sox9* expression were markedly increased in tumor tissues from the *CDX2P-CreER^{T2} Cdx2^{fl/fl} Braf^{CA}* mice (*Figure 3—figure supplement 2A and B*). Expression of *Olfm4, Mmp7* and *Sox9* transcripts was also markedly upregulated in proximal colon tissues from the *CDX2P-CreER^{T2} Cdx2^{fl/fl}* mice. Expression of all of these presumptive stem cell and Wnt pathway genes, was markedly increased in *Apc*-mutant mouse colon tissues (*Figure 3—figure supplement 2A and B*). Consistent with our immunohistochemical studies, *Muc5ac, Anxa10* and *Pdx1* gene expression was increased in the *Cdx2^{−/−} Braf^{V600E}* serrated mouse tumors (*Figure 3—figure supplement 2C*).

## Cooperative interaction of CDX2 silencing and BRAF^{V600E} expression in gene regulation

To address the basis for the major collaborative effects of *Cdx2* inactivation and *Braf^{V600E}* activation in tumorigenesis, we undertook global gene expression analyses of normal colon epithelium, *Cdx2^{−/−}*-mutant, *Braf^{V600E}*-mutant, *Cdx2^{−/−} Braf^{V600E}*-mutant, and *Apc*-mutant colon epithelium. The principal components analysis showed the *Cdx2^{−/−} Braf^{V600E}* mutant epithelium clearly had distinct global patterns of gene expression from that of normal colon epithelium or mutant colon epithelium of the single mutant genotypes (*Figure 5A*). The gene expression patterns in *Cdx2^{−/−}Braf^{V600E}*-mutant epithelium indicated that combined *Cdx2* inactivation and *Braf^{V600E}* activation had very strong cooperative interactions in altering the expression of a large number of genes, with many genes dramatically activated in expression and a smaller collection of genes dramatically down-regulated (*Figure 5B*). Interestingly, notwithstanding the data above showing that the ANXA10 protein was strongly expressed in human serrated morphology CRC and mouse *Cdx2^{−/−} Braf^{V600E}* serrated tumor epithelium, it was still a surprise that *Anxa10* was the top activated gene in the *Cdx2^{−/−} Braf^{V600E}*-mutant epithelium (*Figure 5B*). *Cdx2* loss or *Braf^{V600E}* expression alone had only modest effect on *Anxa10* expression relative to levels in control epithelium, whereas concurrent *Cdx2* and *Braf^{V600E}* defects led to a roughly 100-fold activation of *Anxa10* (*Figure 5B*). Interestingly, the *Pla2g2a* gene, which encodes a secretory phospholipase A2 protein and which is also known as *Mom-1* (*modifier of multiple intestinal neoplasia 1*) for its defined role as a potent genetic modifier of the *Apc^{Min}* intestinal neoplasia phenotype (*Cormier et al., 1997*), was a non-induced gene in *Cdx2^{−/−} Braf^{V600E}*-mutant epithelium compared to its induction in *Cdx2^{−/−}* mutant, *Braf^{V600E}*-mutant, or *Apc*-mutant epithelium (*Figure 5B*). The top pathways for activated gene expression patterns in the *Cdx2^{−/−} Braf^{V600E}*-mutant epithelium were epithelial-mesenchymal transition, KRAS signaling, and hypoxia, and the top pathways for down-regulated gene expression were cholesterol homeostasis, adipogenesis, and xenobiotic metabolism (*Table 2*).

To address the cooperative effects of CDX2 and BRAF function in regulating gene expression in human CRCs, we first compared our mouse gene signature to 212 human CRC samples available in TCGA project data (https://gdac.broadinstitute.org/) and that also have mutation data. Among these samples, we compared the 18 samples that had low CDX2 and BRAF V600E mutations to the 104 samples with high CDX2 that were not BRAF mutant, and selected the genes with *p*-values of <0.01 (two-sample T-test) and fold-changes of >1.3 between the two groups. Then, we computed the intersection of this selection to the similar selection we had performed in our mouse data to ask for significant Cdx2 by Braf interactions. We observed strong association between our mouse data and human CRC data with significant enrichment of genes found 'up' in both data-sets as well as 'down' in both data-sets, and few disagreements between both data-sets (p=$4.5 \times 10^{-49}$, Mantel-Haenszel Chi-Square test of association) (*Figure 5* and *Figure 5—source data 2*). In addition, we also compared our mouse gene signature to expression data for 8 serrated and 29 conventional CRCs obtained from GEO series GSE4045 (called here Finnish data) (*Laiho et al., 2007*). Again, the genes that showed strong cooperative interaction between *Cdx2* loss and *Braf^{V600E}* expression in the mouse colon tumors (either up or down) were found to be significantly enriched in human serrated CRCs vs. conventional CRCs (p=$1.4 \times 10^{-11}$) (*Figure 5—source data 2*). Our findings suggested that the gene signature found in our *Cdx2^{−/−}/Braf^{V600E}* mouse model is highly instructive to identify the serrated subset of human CRCs.

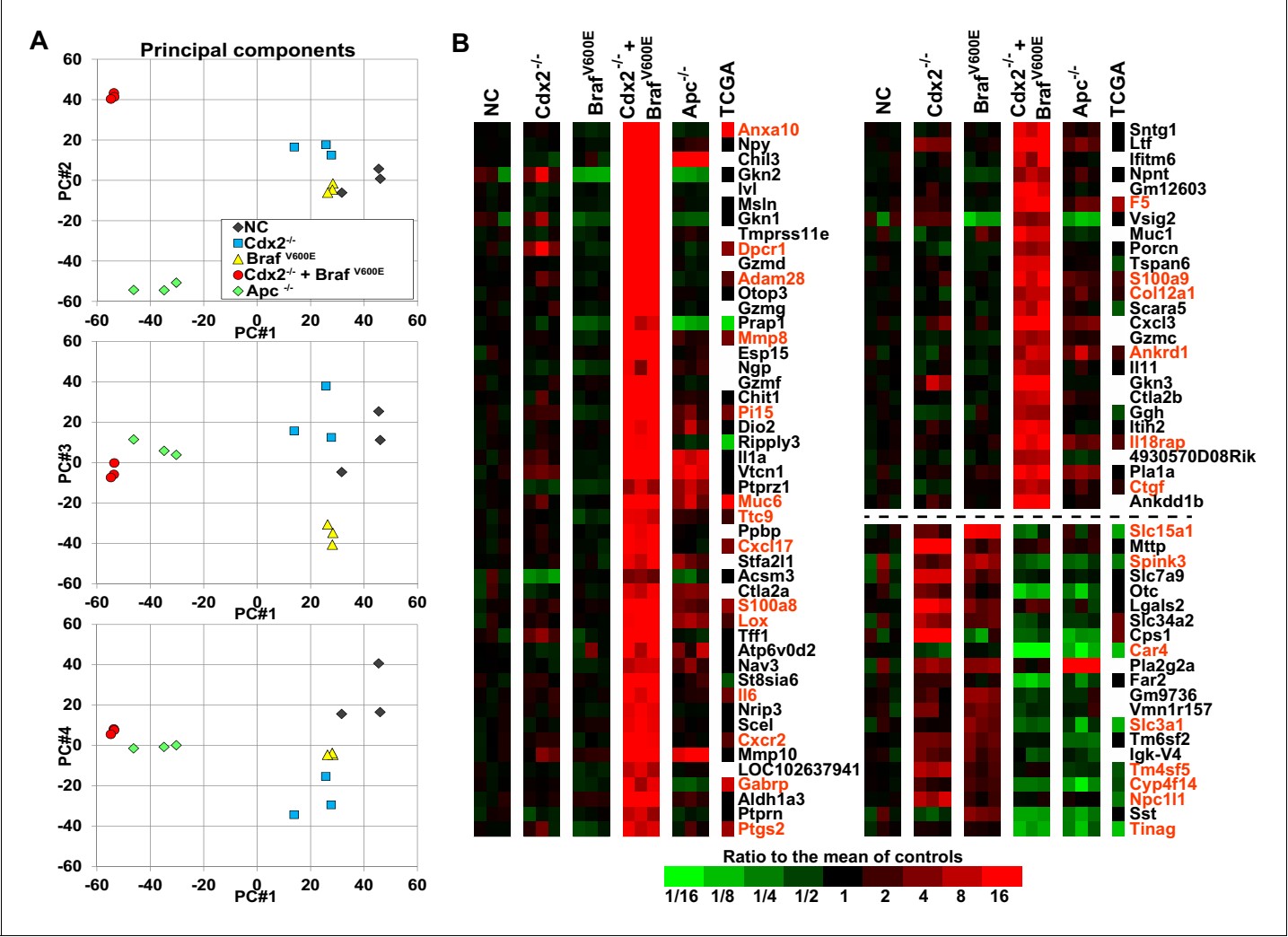

**Figure 5.** Global gene expression analyses show strong cooperative interactions of *Cdx2* inactivation and BRAF[V600E] expression in gene regulation in mouse colon. Global gene expression analyses were performed with RNAs from normal colon epithelium (NC), *Cdx2*[−/−]-mutant, *Braf*[V600E]-mutant, *Cdx2*[−/−] *Braf*[V600E]-mutant, and *Apc*[−/−]-mutant colon epithelium. (**A**) The principal components analysis showed the *Cdx2*[−/−] *Braf*[V600E] mutant epithelium clearly had distinct global patterns of gene expression from that of normal colon epithelium or mutant colon epithelium of the single mutant genotypes. See also *Figure 5–source data 1*. (**B**) Heat map showing the strong cooperative interactions of *Cdx2* inactivation and *Braf*[V600E] activation in altering the expression of many genes. Genes were sorted by their expression based on the degree of interaction between *Cdx2* inactivation and *Braf*[V600E] activation. Genes shown are those that gave p<0.01 for the interaction of *Cdx2* inactivation and *Braf*[V600E] activation (the difference of the Cdx2[−/−] Braf[V600E] vs. Braf[V600E] difference and the Cdx2[−/−] vs. control difference, in log-space) and that also had the anti-logarithm of the difference of differences at least 10 (or smaller than 1/10) in mouse data. Genes below the dashed line show negative interaction between *Cdx2* inactivation and *Braf*[V600E] activation. The heat map also showed the comparison of gene expression between the mouse data and human TCGA CRC data. The 18 CRC samples from human TCGA data that had low CDX2 and V600E mutations were compared to the 104 CRC samples with high CDX2 that were not BRAF mutant, and genes were selected with p-values of <0.01 (two-sample T-test) and fold-changes of >1.3 between the two groups. The genes selected by both human and mouse data sets are marked as red. See also *Figure 5–source data 2,3*.

The following source data is available for figure 5:

**Source data 1.** Raw data for computing the principal components.

**Source data 2.** Comparison of mouse genes up- and down-regulated in Cdx2[−/−] Braf[V600E] tumors to gene signatures from human CRCs with low CDX2expression and BRAF[V600E] mutation (TCGA data) and human serrated carcinomas.

**Source data 3.** Raw data for comparison of mouse genes up- and down-regulated in Cdx2[−/−] Braf[V600E] tumors to gene signatures from human CRCswith low CDX2 expression and BRAF[V600E] mutation (TCGA data) and human serrated carcinomas.

**Table 2.** Top 10 gene sets for lists of genes up- and down-regulated in tumors with $Cdx2^{-/-}$ and $Braf^{V600E}$ mutations.

| Up or down genes | Gene set title | Number of genes on list | Number of those genes we selected | P-value (Fisher Exact) | Observed/ expected | Rank of gene set | Estimated FDR (Q-value) |
|---|---|---|---|---|---|---|---|
| Up | HALLMARK_EPITHELIAL_MESENCHYMAL_TRANSITION | 186 | 69 | 1.0E-28 | 4.57 | 1 | 0.000 |
| Up | HALLMARK_KRAS_SIGNALING_UP | 192 | 41 | 7.3E-09 | 2.63 | 2 | 0.000 |
| Up | HALLMARK_HYPOXIA | 189 | 36 | 1.1E-06 | 2.35 | 3 | 0.000 |
| Up | HALLMARK_COAGULATION | 126 | 27 | 2.4E-06 | 2.64 | 4 | 0.000 |
| Up | HALLMARK_TNFA_SIGNALING_VIA_NFKB | 189 | 35 | 3.2E-06 | 2.28 | 5 | 0.000 |
| Up | HALLMARK_UV_RESPONSE_DN | 141 | 28 | 7.6E-06 | 2.45 | 6 | 0.000 |
| Up | HALLMARK_ANGIOGENESIS | 36 | 12 | 1.5E-05 | 4.11 | 7 | 0.000 |
| Up | HALLMARK_INFLAMMATORY_RESPONSE | 189 | 32 | 5.3E-05 | 2.09 | 8 | 0.000 |
| Up | HALLMARK_IL2_STAT5_SIGNALING | 193 | 30 | 4.2E-04 | 1.92 | 9 | 0.000 |
| Up | HALLMARK_APICAL_JUNCTION | 191 | 29 | 7.6E-04 | 1.87 | 10 | 0.001 |
| Down | HALLMARK_CHOLESTEROL_HOMEOSTASIS | 72 | 17 | 2.7E-09 | 5.86 | 1 | 0.000 |
| Down | HALLMARK_ADIPOGENESIS | 191 | 22 | 9.3E-06 | 2.86 | 2 | 0.000 |
| Down | HALLMARK_XENOBIOTIC_METABOLISM | 194 | 22 | 1.2E-05 | 2.81 | 3 | 0.000 |
| Down | HALLMARK_ESTROGEN_RESPONSE_LATE | 189 | 21 | 2.6E-05 | 2.76 | 4 | 0.000 |
| Down | HALLMARK_MTORC1_SIGNALING | 193 | 21 | 3.5E-05 | 2.70 | 5 | 0.000 |
| Down | HALLMARK_BILE_ACID_METABOLISM | 109 | 14 | 1.2E-04 | 3.19 | 6 | 0.000 |
| Down | HALLMARK_FATTY_ACID_METABOLISM | 149 | 16 | 3.4E-04 | 2.67 | 7 | 0.004 |
| Down | HALLMARK_PEROXISOME | 101 | 12 | 7.4E-04 | 2.95 | 8 | 0.005 |
| Down | HALLMARK_GLYCOLYSIS | 193 | 18 | 8.2E-04 | 2.31 | 9 | 0.004 |
| Down | HALLMARK_ESTROGEN_RESPONSE_EARLY | 190 | 17 | 1.8E-03 | 2.22 | 10 | 0.005 |

Results of enrichment testing the Molecular Signatures Database (MSigDB) v5.1 for the Hallmark collection of 50 gene sets. We selected genes with significant Cdx2 by Braf interactions, asking that p<0.01 and that the fold-change between Cdx2+Braf and Braf was at least 1.3 times larger (or smaller) than the fold-change between Cdx2 and control, which selected 1453 up distict mouse genes and 767 down mouse genes (out of 22326 distinct mouse genes). We mapped to human genes choosing only 1-to-1 best homologs using NCBI's Homologene version 68, which left 1277 up and 634 down distict human genes, out of a total of 15735 genes. We tested our up and down genes separately for over-representation in 50 Hallmark gene sets using one-sided Fisher Exact tests. Shown are the 10 gene sets in the collection that gave the smallest p-values. The last column gives the estimated false discovery rates (Q-values) based on 100 data sets in which the human gene identifiers were randomly permuted.

Source data 1. Results of enrichment testing the Molecular Signatures Database (MSigDB) v5.1 for the hallmark collection of 50 gene sets.

## PDX1 expression in colon tumors in mouse and human

The regulation of PDX1 homeobox transcription factor expression by CDX2 and BRAF in mouse colon tissues was of interest to us, given the role of homeobox proteins in cell-fate specification. Thus, we pursued in-depth immunohistochemical studies of PDX1 expression in selected mouse and human tissue specimens. In the normal mouse gastro-intestinal tissues studied, only gastric antrum epithelium and duodenum epithelium showed positive staining for PDX1, with no staining in epithelium from terminal ileum, cecum, colon and rectum (*Figure 6—figure supplement 1*). In contrast, strong PDX1 staining was seen in colon epithelium from *CDX2P-CreER^{T2} Cdx2^{fl/fl} Braf^{CA} mice* (*Figure 6A*). Strong and diffuse PDX1 staining was also seen in colon epithelium of *CDX2P-CreER^{T2} Cdx2^{fl/fl} mice* (*Figure 6—figure supplement 1*). Activation of PDX1 expression was seen in colon epithelium of *CDX2P-CreER^{T2} Cdx2^{fl/fl} mice* as early as 7 days after TAM injection to target the *Cdx2* alleles (*Figure 6—figure supplement 1*). While strong PDX1 expression was seen in the majority of neoplastic cells in the tumors arising in the *CDX2P-CreER^{T2} Cdx2^{fl/fl} Braf^{CA} mice* (*Figure 6A*), PDX1 expression was largely restricted to apical region epithelial cells in colon epithelium of *CDX2P-CreER^{T2} Cdx2^{fl/fl} mice* (*Figure 6—figure supplement 1*).

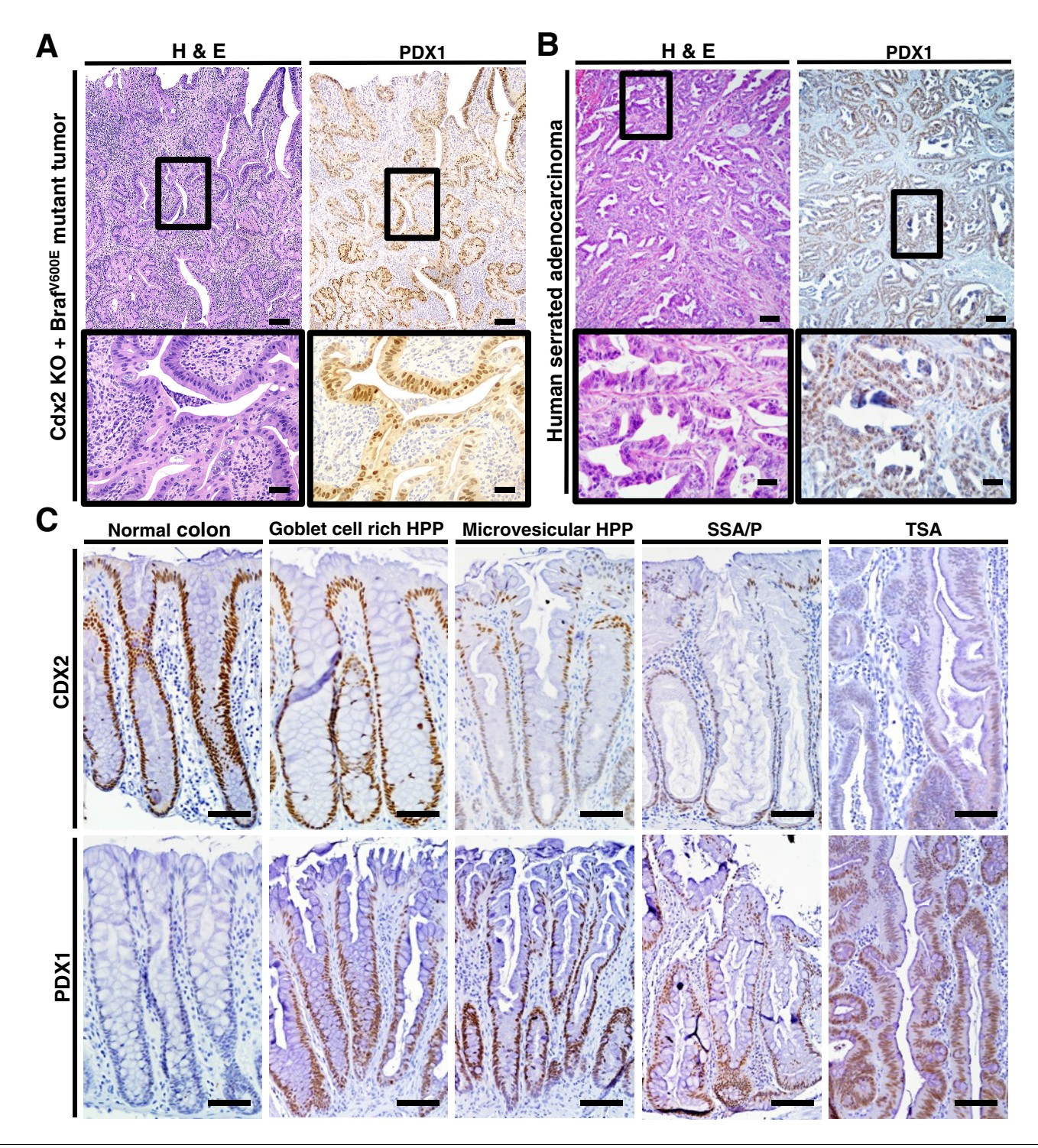

**Figure 6.** PDX1 expression in tumor tissues of *CDX2-CreER^{T2} Cdx2^{fl/fl} Braf^{CA}* mice and human serrated benign tumor tissues. (A) H and E staining (left) and PDX1 immunohistochemical (IHC) staining in the tumor of *CDX2-CreER^{T2} Cdx2^{fl/fl} Braf^{CA}* mice. (B) H and E staining (left) and PDX1 immunohistochemical (IHC) staining in a representative human serrated adenocarcinoma case. Scale bars for panels **A** and **B**, 200 μm for low magnification image (top); 50 μm for high magnification images (bottom). (C) Expression pattern of CDX2 and PDX1 in human serrated benign tumors. (Upper panel) CDX2 expression was retained in Goblet cell rich HPP but significantly decreased in all of the other serrated tumors. (Lower panel) Robust aberrant PDX1 expression was seen in all of the serrated tumors, but was absent in normal colon. Scale bars, 50 μm.

*Figure 6 continued on next page*

*Figure 6 continued*

The following source data and figure supplement are available for figure 6:

Source data 1. Expression of PDX1, CDX2 and ANXA10 in 395 Human CRCs.
Figure supplement 1. Distribution of PDX1 expression in mouse gastrointestinal tract.

We then assessed the 36 serrated morphology human CRC cases, finding that 30 (83.3%) had strong PDX1 expression, including all 26 of the cases with CDX2 loss/reduction and/or BRAF$^{T1799A}$ mutations (*Figure 1* and *Figure 6B*). To assess whether reduction or loss of CDX2 expression and ectopic PDX1 expression were also features of benign serrated colorectal lesions, we studied CDX2 and PDX1 expression in four goblet cell-rich histology HPPs, ten microvesicular histology HPPs, six SSAs/SSPs, and four TSAs. While CDX2 expression was retained in all goblet cell-rich histology HPPs, all 20 of the other benign serrated lesions studied had greatly reduced or absent CDX2 expression (*Figure 6C*). Moreover, induction of PDX1 expression was seen in all benign serrated lesion types (*Figure 6C*) and in all 24 benign lesions studied.

Because of the robust expression of PDX1 in all 24 of the serrated benign lesions studied and in 83.3% of the serrated morphology CRCs studied, we sought to assess the expression of PDX1 as well as CDX2 and ANXA10 expression in a large collection of CRCs not selected for any specific morphologic features. We found PDX1 was expressed in 132 (33%) of the 395 CRCs studied (*Figure 6—source data 1*), a percentage which is curiously not far from the upper estimate that 30% of CRCs may arise through a serrated pathway. Strong staining for CDX2 was seen in 287 (73%) of the 395 CRCs, with reduced but detectable CDX2 staining in 73 (18%) of the 395 cases and loss of CDX2 expression in 35 (9%) of the 395 cases (*Figure 6—source data 1*). ANXA10 expression, which was seen in 24 (67%) of the 36 human serrated CRCs, was only expressed in 26 (7%) of the 395 CRCs. These findings on ANXA10 expression, based on our data showing that only about 2/3 of serrated morphology CRCs express ANXA10, are not inconsistent with the published literature indicating that about 8–10% of all CRCs have serrated morphology. Because the 395 CRCs were represented as very limited tumor regions in the tissue microarray format, we could not reliably assess relationships of PDX1, CDX2 and ANXA10 staining in these CRCs relative to serrated morphology features.

## Serrated glandular morphology and gene expression in organoids derived from CDX2$^{Null}$/BRAF$^{V600E}$-mutant colon epithelium

To further assess how concurrent *Cdx2* inactivation and BRAF$^{V600E}$ expression alters colon epithelial cell phenotypes, we studied organoids derived from normal and mutant mouse colon epithelium. Normal mouse colon epithelium-derived organoids formed uniform glandular structures in the presence of media supplemented with Wnt3a, R-spondin, noggin, epidermal growth factor (EGF), and hepatocyte growth factor (HGF) (*Figure 7*). The cells in the normal epithelium-derived organoids lacked dysplasia and resembled normal columnar colon epithelial cells. *Apc*-mutant (*Apc$^{-/-}$*) mouse colon epithelium formed organoids in the absence of Wnt3a and R-spondin supplementation, as expected for cells with bi-allelic *Apc* mutations and constitutive activation of $\beta$-catenin/TCF transcription. The *Apc$^{-/-}$* epithelium-derived organoids were larger than normal epithelium-derived organoids, and the cells showed dysplasia, with larger and more irregularly shaped nuclei containing open chromatin and a loss of the normal basal polarity of the nuclei (*Figure 7*). Organoids derived from Cdx2$^{Null}$ mouse colon epithelium were grown in the presence of the full growth factor supplementation, and the organoids showed minimal if any alterations in morphology relative to organoids derived from normal colon epithelium (*Figure 7*). Organoids derived from CDX2$^{Null}$/BRAF$^{V600E}$-mutant epithelium could be grown without EGF supplementation of the media, due presumably to constitutive BRAF→MEK1/2→ERK1/2 MAP kinase signaling driven by the BRAF$^{V600E}$ oncoprotein kinase. Initially, roughly 25% of the organoids derived from the CDX2$^{Null}$/BRAF$^{V600E}$-mutant epithelium showed serrated glandular structures, with invaginations of epithelial cells into the organoid lumen, and epithelial cells with eosinophilic or clear and abundant cytoplasm, vesicular nuclei with distinct nucleoli, and loss of basal polarity of the cells, closely resembling mouse and human

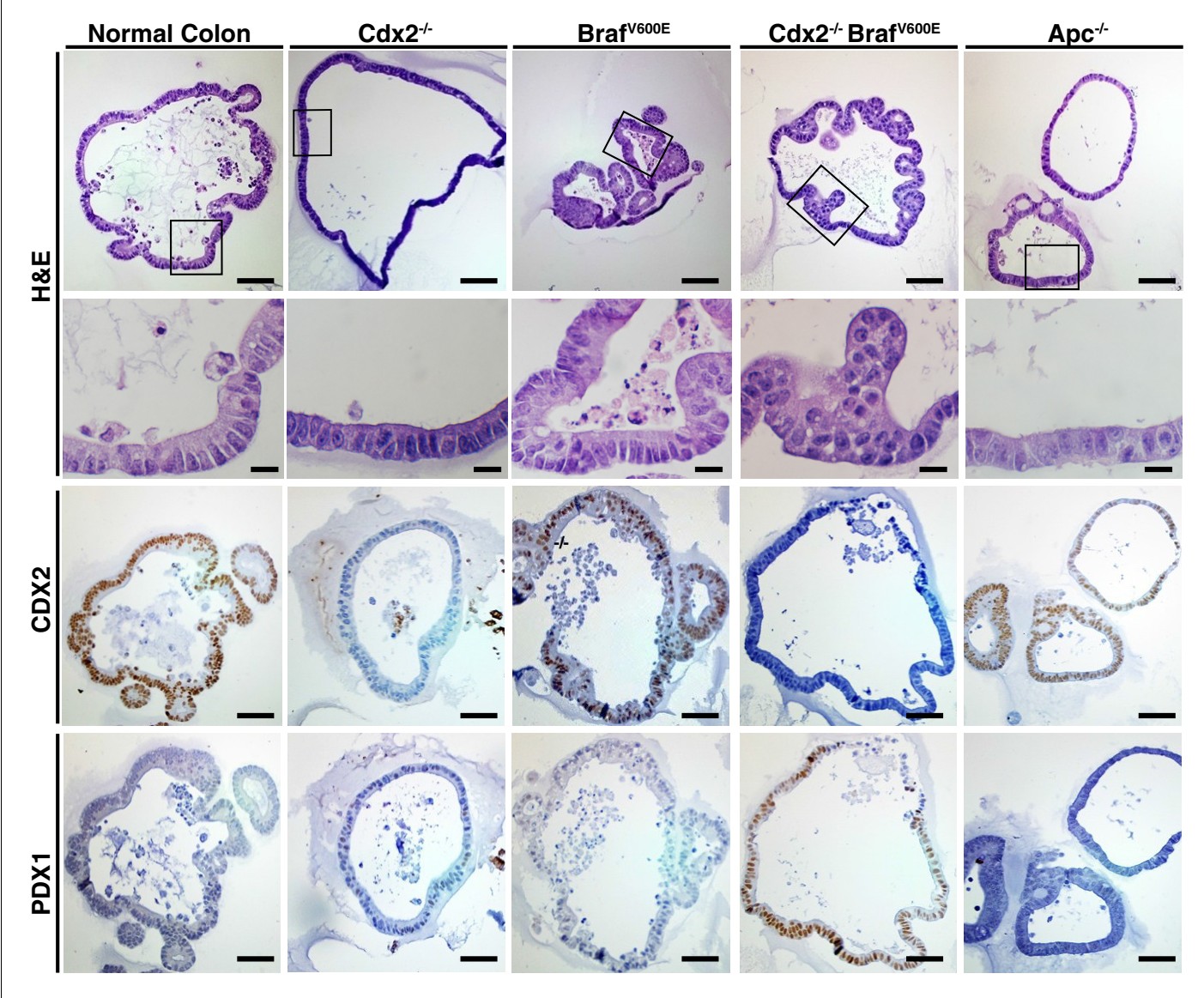

**Figure 7.** Organoids derived from *Cdx2*$^{-/-}$ and *Braf*$^{V600E}$-mutant colon epithelium showed serrated glandular morphology and PDX1 expression. Organoids were generated from proximal colon of wild-type, *CDX2P-CreER*$^{T2}$ *Apc*$^{fl/fl}$, *CDX2P-CreER*$^{T2}$ *Cdx2*$^{fl/fl}$, *CDX2P-CreER*$^{T2}$ *Braf*$^{CA}$, and *CDX2P-CreER*$^{T2}$ *Cdx2*$^{fl/fl}$ *Braf*$^{CA}$ mice after TAM injection. Representative organoids from each mouse were shown for H and E (upper panels), and immunohistochemical staining for CDX2 (middle panels) and PDX1 (lower panels). Scale bars, 200 μm for low magnification; 50 μm for high magnification (for H and E).

epithelium with serrated morphology (*Figure 7*). CDX2 expression was seen in all organoids where *Cdx2* gene function remained intact (*Figure 7*). Strong and homogeneous PDX1 expression was found only in the CDX2$^{Null}$/BRAF$^{V600E}$-mutant epithelium. After three-four in vitro passages of the CDX2$^{Null}$/BRAF$^{V600E}$-mutant organoids, we observed a significant decrease in the percentage of organoids displaying serrated morphology, suggesting that serrated morphologic appearance as a phenotypic trait may not be stable in some neoplastic cell populations over time. To further explore the usefulness of the CDX2$^{Null}$/BRAF$^{V600E}$-mutant organoids as a model system, we studied the gene expression by q-RT PCR for the top 15 genes showing positive interaction of CDX2 silencing and BRAF$^{V600E}$ expression in mouse colon tissues and five more 'up' genes that are shared by our mouse data-set and one of the two human CRC data sets – the TCGA data-set or the Finish data-set (*Figure 5—source data 3*, marked as red). Interestingly, only 4 out of the 20 genes up-regulated in the

in vivo setting (i.e., *Npy*, *Gkn2*, *Ivl* and *Tmprss11e*) showed significant cooperative interaction between CDX2 silencing and BRAF$^{V600E}$ expression in the organoid model system (*Figure 8A and C*). Lower expression of *Anxa10* was found in the CDX2$^{Null}$/BRAF$^{V600E}$-mutant organoids compared to the normal control or *Braf*$^{VE}$ organoids (*Figure 8A and C*). The expression of *Npy*, *Gkn2*, *Ivl* and *Tmprss11e* as well as *Anxa10* in CDX2$^{Null}$/BRAF$^{V600E}$-mutant organoids was markedly inhibited by MEK1/2-inhibitor (AZD6244) treatment. The effectiveness of AZD6244 treatment was verified by its inhibitory effect on expression of MAPK target genes, *Dusp6* and *Fos*, in all the organoids treated (*Figure 8B and C*) and was also confirmed by its ability to significantly reduce the level of phospho-ERK (*Figure 8—figure supplement 1*).

## Discussion

Studies of the molecular pathogenesis of CRC have often focused on a conventional adenoma-carcinoma progression model (*Fearon, 2011*; *Fearon and Vogelstein, 1990*). But, clinical, pathological, and molecular data suggest perhaps upwards of 20–30% of CRCs may arise from benign lesions with serrated glands, rather than conventional adenomas (*Bettington et al., 2013*; *Jass, 2007a*; *Langner, 2015*). One challenge in studying the pathogenesis of the CRCs that arise from serrated precursors is there are no known biomarkers that can definitively distinguish CRCs that arose from serrated lesions from CRCs arising from conventional adenomas (*Bettington et al., 2013*; *Jass, 2007a*; *Langner, 2015*). Another issue is that while an estimated 20–30% of CRCs may arise from a serrated precursor lesion, at diagnosis, only about 8–10% of CRCs manifest serrated morphological features (*Bettington et al., 2013*). In the work presented here, we have provided new insights into the molecular pathogenesis of serrated morphology CRCs. We have described a novel mouse tumor model based on two signature molecular lesions present in many human serrated morphology CRCs, and we provide evidence that the mouse tumors manifest significant phenotypic similarities to human serrated morphology CRCs. We have also highlighted new expression markers that may be useful for studying further the serrated precursor-CRC relationship.

We found marked loss or reduction of the CDX2 homeobox transcription factor and *BRAF* somatic mutations were both prevalent in serrated morphology CRCs, with concurrent CDX2 loss and *BRAF*$^{T1799A}$ mutation in 15 of the 36 (42%) cases. To address the potential cooperating roles of CDX2 loss and BRAF$^{V600E}$ expression in human serrated morphology CRCs, we studied the consequences of conditional inactivation of *Cdx2* alleles, expression of a BRAF$^{V600E}$ mutant protein, or concurrent *Cdx2* inactivation and BRAF$^{V600E}$ mutant protein expression in mouse distal intestinal epithelium. We found concurrent *Cdx2* inactivation and BRAF$^{V600E}$ expression had dramatic effects on mouse survival relative to either defect alone, with a median survival for the *CDX2-CreER*$^{T2}$ *Cdx2*$^{fl/fl}$ *Braf*$^{CA}$ mice of only 103 days after conditional gene targeting. The mice developed large tumors in the terminal ileum, cecum and/or proximal colon, and the tumors displayed serrated histological features. A subset of the tumors arising in each *CDX2-CreER*$^{T2}$ *Cdx2*$^{fl/fl}$ *Braf*$^{CA}$ mouse had clear evidence of invasion (carcinoma). Molecular analyses of the mouse tumors revealed that concurrent *Cdx2* loss and BRAF$^{V600E}$ expression had major cooperating effects in altering gene expression. Cooperating effects of CDX2 and BRAF function in modulating expression of selected genes could also be seen in CDX2$^{Null}$/BRAF$^{V600E}$-mutant organoids. However, the organoid system did not model the cooperative effects of CDX2 and BRAF defects for the vast majority of the genes found to be upregulated in CDX2$^{Null}$/BRAF$^{V600E}$-mutant epithelium in vivo, raising some questions about the utility of in vitro organoid model for recapitulating cooperation between CDX2 and BRAF. Human serrated morphology CRCs and mouse *Cdx2*$^{-/-}$ *Braf*$^{V600E}$-mutant tumor epithelium aberrantly expressed a number of gastric epithelial markers, including ANXA10, MUC5AC, and PDX1.

Prior studies have emphasized the view that SSAs/SSPs, especially SSAs/SSPs arising in the proximal colon, are the serrated tumors with the greatest risk of progression to CRC (*Bettington et al., 2013*; *Jass, 2007a*; *Langner, 2015*). Besides the *BRAF*$^{T1799A}$ mutations that are often present in SSAs/SSPs, it has been argued that CRCs arising from proximal colon SSAs/SSPs will often manifest the CIMP-H and MSI-H phenotypes, with the CIMP-H phenotype reflecting hypermethylation of selected promoters, including that for the *MLH1* gene, leading to the MSI-H phenotype (*Bettington et al., 2013*; *Jass, 2007a*; *Langner, 2015*). Some prior studies have shown CRCs with somatic *BRAF*$^{T1799A}$ mutations, and that were not selected by morphological appearance, more often arise in the proximal colon than the distal colon or rectum, and these particular CRCs more

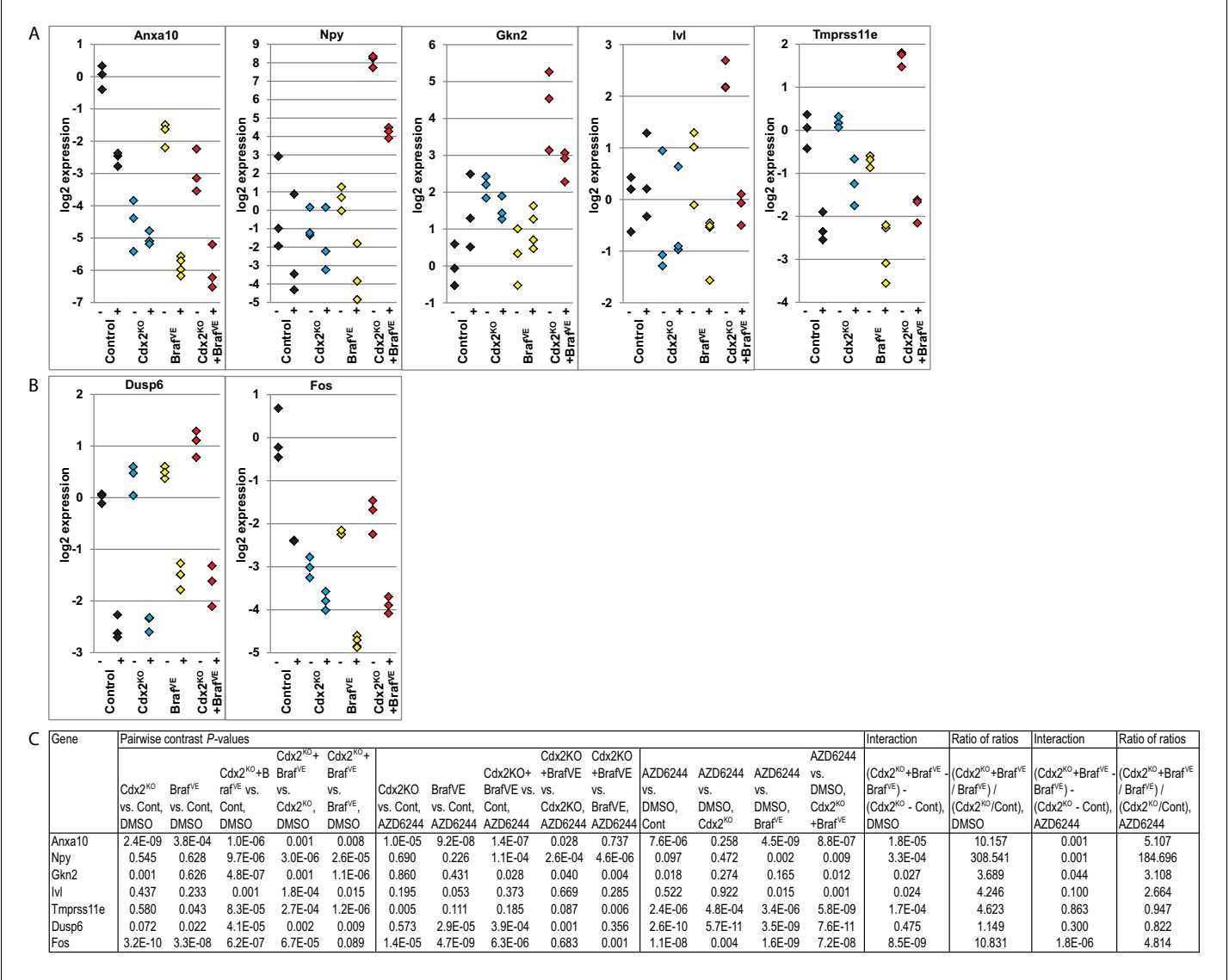

**Figure 8.** Cooperative effects of CDX2 and BRAF function in regulating gene expression in organoids derived from *Cdx2⁻/⁻* and *Braf^V600E*-mutant colon epithelium. Organoids were generated from epithelium at proximal colon-cecum junction of wild-type (Control), *CDX2P-CreER^T2 Cdx2^fl/fl* (*Cdx2^KO*), *CDX2P-CreER^T2 Braf^CA* (*Braf^VE*), and *CDX2P-CreER^T2 Cdx2^fl/fl Braf^CA* (*Cdx2^KO Braf^VE*) mice after 7 days following two daily doses of TAM (150 mg/kg) injection. The organoids were plated in Matrigel for 3 days, and then were treated with DMSO (−) or 6 µM of MEK1/2 inhibitor, AZD6244 (+), daily for two consecutive days before harvest for RNA extraction. Gene expression was assessed by qRT-PCR, with normalization to *β-actin* expression. Independent RNA samples were prepared from triplicate treatments for each treatment group. (A) Expression of *Anxa10*, and the genes showing cooperative interactions of CDX2 silencing and BRAF^V600E expression in both mouse tissues and organoid system. (B) Expression of the MAPK target genes, *Dusp6* and *Fos*, to show the effectiveness of AZD6244 treatment. (C) We fit a one-way ANOVA model with terms for the means of four groups of genotypes (for both DMSO and AZD6244 treatments) to log-transformed gene expression data, and test differences in means for pairs of groups as well as *Braf* by *Cdx2* interactions by testing if the average difference (of log-transformed data) between Cdx2^KO Braf^VE samples and Braf^VE samples was larger or smaller than the difference between Cdx2^KO and control samples, and called the anti-logarithm of this difference-of-differences the ratio of ratios. In addition, we also tested the effects of AZD6244 on gene expression within each genotype groups. See also *Figure 8—source data 1*.

The following source data and figure supplement are available for figure 8:

**Source data 1.** Raw data for qRT-PCR analysis for organoids shown in *Figure 8*.

**Figure supplement 1.** MAPK activity in organoids was significantly blocked by the treatment with MEK inhibitor, AZD6244.

often manifest the CIMP-H and MSI-H phenotypes than the bulk of CRCs(*Bettington et al., 2013*; *Jass, 2007a*). However, in contrast to the notion that CRCs arising from serrated precursors are nearly always or even predominantly right-sided lesions with CIMP-H and MSI-H phenotypes, 17 of the 36 serrated morphology CRCs that we studied were rectal lesions and four were distal colon lesions, with only 15 of the 36 serrated morphology CRCs arising in the proximal colon. We did observe the CIMP-H phenotype more frequently in the human serrated morphology CRCs with CDX2 loss and $BRAF^{T1799A}$ mutations (i.e., CIMP-H in 10 of 15 such cases) than in the CRCs lacking concurrent CDX2 loss and $BRAF^{T1799A}$ mutations (i.e., CIMP-H in only 3 of 21 such cases). However, our 10 CIMP-H cases with concurrent CDX2 loss and $BRAF^{T1799A}$ mutation included four cases arising in the rectum and only 2 of 10 CIMP-H cases displayed the MSI-H phenotype and had lost MLH-1 protein expression. Hence, our data imply that a potentially significant fraction of serrated morphology CRCs may arise from precursor lesions other than proximal colon SSAs/SSPs. Also, while CDX2-null/low and $BRAF^{V600E}$-mutant serrated morphology CRCs often manifested the CIMP-H phenotype, most serrated morphology CRCs lacking concurrent CDX2 and $BRAF^{V600E}$ defects did not manifest the CIMP-H phenotype. In addition, the MSI-H phenotype was not significantly enriched in our 36 serrated morphology CRCs as a group (5 of 36 cases – 14%) relative to the established prevalence of 15% in CRCs in general (*Bettington et al., 2013*; *Cancer Genome Atlas Network, 2012*; *Fearon, 2011*). Consistent with our findings, a prior study reported that the MSI-H phenotype was only seen in 16% of serrated morphology CRCs (*García-Solano et al., 2010*).

The mouse model of serrated intestinal tumorigenesis, including carcinomas, that we describe here, based on concurrent *Cdx2* inactivation and $BRAF^{V600E}$ expression in distal mouse intestinal epithelium, is the first serrated mouse colon cancer model to our knowledge to be based on two signature defects common to a significant subset of human serrated morphology CRCs. Prior serrated tumorigenesis models using conditional activation of a mutant *Kras* allele in the mouse intestinal tract together with concurrent $p16^{Ink4a}/Arf$ or *Pten* TSG inactivation(*Bennecke et al., 2010*; *Davies et al., 2014*), can promote outgrowth of dysplastic lesions and occasionally progression of some lesions to carcinoma. However, our data suggest potentially uncertain relevance of *Kras*-mutation-based mouse models with respect to serrated morphology human CRCs, as only seven of the 36 (20%) serrated morphology CRCs that we studied had *KRAS* mutations and only two of the seven *KRAS*-mutant serrated CRCs had lost $p16^{INK4A}$ expression. Besides the modest-moderate frequency of *KRAS* mutations in human serrated CRCs, the low overall frequency of *PTEN* mutations in human primary CRCs raises some questions about the relevance of mouse models combining *Kras* activation and *Pten* inactivation to model serrated CRC. A prior mouse model has been reported where $Braf^{V600E}$ conditional activation in small intestine in concert with $p16^{Ink4a}$ inactivation led to tumors with some serrated morphological features(*Carragher et al., 2010*). While $p16^{INK4A}$ expression loss is common in $BRAF^{V600E}$-mutant human serrated CRCs, the prior $BRAF^{V600E}$-expressing and $p16^{Ink4a}$-mutant mouse model developed tumors in many tissues besides the intestinal tract (e.g., liver, lung, stomach), limiting its utility. In another model of serrated pathway tumorigenesis, Rad *et al* showed that activation of $Braf^{V600E}$ specifically in mouse intestinal tissues instigated hyperplasia and dysplasia with serrated features, and a very small subset of the tumors progressed over time to invasive carcinomas (*Rad et al., 2013*). However, the $BRAF^{V600E}$-induced dysplastic lesions in this model showed features akin to human traditional serrated adenomas (TSA), which the authors reasoned might be due in part to the predominant location of the tumors arising in mouse small intestine. Our human serrated CRC data highlight additional gene defects that may cooperate with *Cdx2* inactivation and $BRAF^{V600E}$ expression in our mouse model, such as $p16^{Ink4a}$ and/or *Trp53* mutations, and restricted gene targeting to the terminal ileum, cecum and colon as permitted with the $CDX2$-$CreER^{T2}$ transgene may be useful in future studies to determine how additional gene alterations collaborate with the CDX2 and $BRAF^{V600E}$ defects in tumor progression and metastasis.

Among a number of poorly understood issues in the serrated precursor-CRC field is how to reconcile the estimate that upwards of 20–30% of CRCs may arise from serrated precursor lesions, yet only 8–10% of CRCs manifest serrated morphology at diagnosis. One possible explanation is that during progression to CRC the serrated morphological features present in a precursor cell population may be lost, due in part to the initiating gene lesions and additional genetic and epigenetic alterations, changes in the tumor microenvironment, or other mechanisms. Consistent with the view that phenotypic drift during neoplastic progression might contribute to why many fewer CRCs manifest serrated morphology at diagnosis relative to the fraction of CRCs estimated to arise from

serrated precursors, we found the propensity of the CDX2$^{Null}$/BRAF$^{V600E}$-mutant mouse organoids to manifest serrated morphological features in culture decreased substantially over relatively few passages. Another possible issue of interest for the field is which biomarkers might identify the CRCs arising from serrated precursor lesions. Our data on ANXA10 and PDX1 expression may be interesting in this regard. Our findings indicate ANXA10 may not be expressed in all serrated morphology CRCs, with ANXA10 most often expressed in those cases with CDX2 loss and/or BRAF$^{T1799A}$ mutations. On the other hand, PDX1 expression was seen in all benign serrated lesions studied and 83.3% of the 36 serrated morphology CRCs as well as 33% of nearly 400 CRCs selected without regard to specific morphological features. Further studies to define how PDX1 expression and perhaps ANXA10 expression might be used together with other biomarkers to identify the CRCs that arose from serrated precursor lesions may have possible diagnostic and prognostic relevance. Yet another unresolved issue for the serrated precursor-CRC field, and potentially one of greater significance for colorectal cancer screening recommendations in the general population in the future, is whether a potentially significant fraction of human CRCs arise from benign serrated lesions that are currently presumed to be of low malignant potential, such as HPPs in the distal colon and rectum, with or without BRAF$^{T1799A}$mutations, or proximal colon serrated lesions with KRAS mutations. The basis for this proposal is that our studies indicate that many human serrated morphology CRCs may arise in the distal colon and rectum, about half of the rectal and distal colon lesions lack BRAF$^{T1799A}$ mutations, and five of the seven KRAS-mutant serrated CRCs we studied arose in the proximal colon. Further in-depth studies of benign and malignant serrated morphology colorectal lesions in man and longitudinal studies of tumor features and progression in our mouse model may help to inform some of these and other issues about the pathogenesis and clinical features and significance of serrated benign lesions and serrated CRCs.

## Materials and methods

### Human serrated tumor tissues and histological interpretation

The diagnosis of serrated morphology CRC was based on established histopathological criteria (*Mäkinen, 2007*; *Tuppurainen et al., 2005*; *Bateman, 2014*). Serrated morphology CRC was characterized by evident epithelial serrations composed only of epithelium or epithelium and basement membrane material, clear or eosinophilic and often abundant cytoplasm, vesicular nuclei, absence of or less than 10% necrosis of the total surface area, mucin production, and the presence of serrations and eosinophilic cell globules and rod-like structures floating freely in the mucus. The diagnostic sub-classification of the 36 serrated CRCs studied here was based on previously proposed criteria from Snover and Torlakovic *et al* (*Snover, 2011*; *Torlakovic et al., 2003*). Serrated CRCs of each of the three known histological sub-groups, arising in Japanese patients, were included, with the vast majority of cases representing the serrated subtype (n = 30) and only a minority reflecting the mucinous (n = 4) or trabecular (n = 2) subtypes of serrated CRC. The 36 serrated morphology CRCs were studied in accordance with the Ethical Guidelines for Human Genome/Gene Research enacted by the Japanese Government. Hematoxylin- and eosin-stained tissue sections of the human serrated morphology CRCs were evaluated by three board-certified surgical pathologists with expertise in gastrointestinal cancer diagnosis (N.S., K.S., W.Y.). We also studied tissue sections of four Goblet cell rich HPP, 10 Microvesicular HPP, 6 SSA/P, and 4 TSA cases obtained from the University of Michigan tissue procurement service through an Institutional Review Board-approved protocol.

### Analysis of BRAF and KRAS mutations

DNA was extracted from formalin-fixed, paraffin-embedded serrated CRC specimens. Areas in which tumor cells were most dense were delineated by light microscopic analysis of tissue slides. The corresponding areas were marked on 10 serial, unstained tissue slides; the marked areas were then manually scraped from the glass slides. DNA was extracted using QIAamp DNA FFPE Tissue Kit (QIAGEN, Valencia, CA). BRAF codon 600 flanking sequences were amplified by PCR. PCR amplifications were performed with GoTaq Green Master Mix (Promega, Madison, WI) following the standard 3-temperature PCR protocol, with denaturing at 94°C, annealing at 55°C, and extension at 72°C. The PCR products were extracted using QIAquick Gel Extraction Kit (QIAGEN) and submitted for direct Sanger sequencing. The obtained sequences were analyzed and aligned with BRAF reference

sequence, NM_004333.4. Cases apparently negative for $BRAF^{T1799A}$ were evaluated for alternative mutations in *KRAS* codons 12, 13, and 61 by PCR amplification and direct Sanger sequencing of PCR amplification products. Primer sequences were: *KRAS* codon 12/13, 5′- gcctgctgaaaatgactgaat −3′ and 5′- ggtcctgcaccagtaatatgc −3′; *KRAS* codon 61, 5′- ccagactgtgtttctccttc −3′ and 5′-aaa-gaaagccctccccagt-3′; *BRAF* codon 600, 5′- ccacaaaatggatccagaca −3′ and 5′- cctaaactcttca-taatgcttgctc −3′.

## Mice

$Braf^{CA}$ (aka $Braf^{tm1Mmcm}$) mice carry a conditional allele of *Braf* that encodes normal BRAF prior to Cre-mediated recombination, after which it encodes a mouse-human hybrid V600E oncogenic protein, were previously described (*Dankort et al., 2007*). Regardless of species of origin, we refer to this mutationally activated form of BRAF as $BRAF^{V600E}$ and the $Braf^{CA}$ allele after Cre-mediated recombination as $Braf^{V600E}$ for the sake of clarity. $CDX2P-CreER^{T2}$ transgenic mice (*Feng et al., 2013*) were intercrossed with mice homozygous for a *Cdx2* targeted allele ($Cdx2^{fl/fl}$) (*Blij et al., 2012*) and/or the $Braf^{CA}$ mice. Cre-mediated conversion of the $Braf^{CA}$ allele to encode $BRAF^{V600E}$ and the deletion of *Cdx2* were assessed by PCR as previously described (*Blij et al., 2012*; *Dankort et al., 2007*; *Feng et al., 2013*). $Apc^{fl}$ mice have been previously described (*Shibata et al., 1997*). Littermates differing in specific genotypes were used for the survival work and all tissue and molecular work. Animal husbandry and experimental procedures were carried out under approval from the University of Michigan's Institutional Animal Care and Use Committee (PRO00005075) and according to Michigan state and US federal regulations. All the mice were housed in specific-pathogen free (SPF) conditions. After weaning, rodent 5001 chow and automatically supplied water were provided *ad libitum* to mice.

## Tamoxifen (TAM) treatment

Adult mice (2–3 months of age) carrying the $CDX2P-CreER^{T2}$ transgene were injected intraperitoneally with TAM (Sigma-Aldrich, St. Louis, MO) dissolved in corn oil (Sigma-Aldrich). For two daily TAM doses, we used 150 mg/kg weight per dose. Animals were euthanized and analyzed at various time points after the final injection given.

## Establishment of colonic organoids

Organoids were derived from proximal colon of the following mice: (1) wild-type mice (6-week age); (2) $CDX2P-CreER^{T2}$ $Apc^{fl/fl}$ mice (24 days after TAM injection); (3) $CDX2P-CreER^{T2}$ $Cdx2^{fl/fl}$ mice (three weeks after TAM injection); (4) $CDX2P-CreER^{T2}$ $Braf^{CA}$ mice (3 months after TAM injection); (5) $CDX2P-CreER^{T2}$ $Cdx2^{fl/fl}$ $Braf^{CA}$ mice (4 months after TAM injection). Mice were first treated with TAM to induce gene targeting and/or tumor formation, and then colonic organoids were generated and propagated using previously described 'TMDU protocol' (*Yui et al., 2012*) with minor modifications. The proximal colon and tumor lesions in the proximal colon were removed, minced into small pieces, and suspended in 12.5 ml of DMEM (Life Technologies Corporation, Grand Island, NY) supplemented with 100 U/ml penicillin (Life Technologies Corporation), 100 µg/ml streptomycin (Life Technologies Corporation), 50 µg/ml gentamicin (Life Technologies Corporation) and 1% FBS (complete DMEM), to which 15 mM EDTA were added. The mixture was shaken for 1 hr at 4°C. The released crypts were washed extensively, pelleted, and resuspended in 200 µl of the Matrigel (Corning, Bedford, MA) and placed in 3 wells of 24-well plates. After polymerization, 500 µl of 'TMDU' medium, which has advanced DMEM/F12 (Life Technologies Corporation) containing 2 mM L-Glutamine (Life Technologies Corporation), 10% Wnt3A (*Barker et al., 2010*), 10% R-spondin2 (*Bell et al., 2008*) conditioned-media and 100 ng/ml Noggin (Peprotech, Rocky Hill, NJ), 20 ng/ml mouse EGF (Life Technologies Corporation), 50 ng/ml mouse HGF (R and D Systems, Minneapolis, MN), and 10 µM Y-27632 (R and D Systems), was added to each well for organoids from wild-type mice and $CDX2P-CreER^{T2}$ $Cdx2^{fl/fl}$ mice. We used TMDU medium minus mEGF for the organoids from $CDX2P-CreER^{T2}$ $Braf^{V600E}$ and $CDX2P-CreER^{T2}$ $Cdx2^{fl/fl}$ $Braf^{V600E}$ mice, and used TMDU medium with Wnt3a and R-spondin2 conditional media replaced by advanced DMEM/F12 for organoids from $CDX2P-CreER^{T2}$ $Apc^{fl/fl}$ mice. The medium was changed every 2–3 days. The organoids were split at 1:3 ratios every 7–8 days. For gene expression study, organoids were generated from epithelium at proximal colon-cecum junction of wild-type (Control), $CDX2P-CreER^{T2}$ $Cdx2^{fl/fl}$

($Cdx2^{KO}$), CDX2P-CreER$^{T2}$ Braf$^{CA}$ (Braf$^{VE}$), and CDX2P-CreER$^{T2}$ Cdx2$^{fl/fl}$ Braf$^{CA}$ ($Cdx2^{KO}$Braf$^{VE}$) mice after 7 days following two daily doses of TAM (150 mg/kg) injection. The organoids were plated in Matrigel for 3 days, and then were treated with DMSO or 6 μM of MEK1/2 inhibitor, AZD6244 (Cayman Chemical Company, Ann Arbor, MI), daily for two consecutive days before harvest for RNA extraction.

## Immunohistochemistry (IHC)

Mouse tissues were prepared for paraffin embedding, as described previously (*Feng et al., 2013*). For assessment of cell proliferation, mice were pulsed with 5-bromo-2-deoxyuridine (BrdU; Sigma-Aldrich) for 1 hr before the mice were euthanized. Sections of paraffin-embedded human or mouse tissues were subjected to immunohistochemical (IHC) analysis as described (*Feng et al., 2011*). Primary antibodies used for IHC analysis with sections of paraffin-embedded mouse and human tissues are listed in *Supplementary file 1*. For BrdU staining, tissue sections were treated with 2N HCl at 37°C for 30 min after performing antigen retrieval with citrate buffer (pH 6.0, BioGenex, Fremont, CA). All IHC analyses on human serrated CRC tissues and the mouse tissues were evaluated by three board-certified surgical pathologists with expertise in gastrointestinal cancer diagnosis (N.S., K.S., W.Y.). All of the IHC staining was classified according to the percentage of stained cells. With the exception of CDX2 staining in human CRC specimens, expression of all markers in the human and mouse tissue samples were considered to be negative if <10% of neoplastic cells were observed to be stained, and positive when >10% of neoplastic cells were stained.

## MSI and CpG island methylation analyses of human serrated CRCs

We selected five markers for investigation of MSI status in the human serrated CRCs: BAT-25, BAT-26, NR-21, NR-22, NR-24. One primer in each pair was labeled with one of the fluorescent markers FAM, HEX or NED. The five mononucleotide repeat tracts were co-amplified in one multiplex PCR containing 1 mol/L of each primer, 200 mol/L dNTP, 1.5 mmol MgCl$_2$ and 0.75 U Taq DNA polymerase. The pentaplex PCR was performed under the following conditions: denaturation at 94°C for 5 min, 35 cycles of denaturation at 94°C for 30 s, annealing at 55°C for 30 s, and extension at 72°C for 30 s. This was followed by an extension step at 72°C for 7 min (*Suraweera et al., 2002*). The PCR products were analyzed by ABI 3730XL Genetic Analyzers (Applied Biosystems, Foster City, CA) and Gene Marker-HD software (Softgenetics, State College, PA) was used to calculate the size of each fluorescent PCR product. For the CpG island methylation analysis, sodium bisulfite treatment of genomic DNA and subsequent real-time PCR (MethyLight) were validated and done as previously described (*Weisenberger et al., 2005*). We quantified DNA methylation at five CIMP-specific promoters: CACNA1G, IGF2, NEUROG1, RUNX3 and SOCS1. CIMP-high was defined as the presence of more than 3 of 5 methylated promoters, CIMP-low as the presence of 0 to 2 of 5 methylated promoters, according to the previously established criteria (*Weisenberger et al., 2006*).

## MSI analysis study and bisulfite sequencing on p16 promoter in mouse tissues

Microsatellite analysis of mouse tumors was done as described using five microsatellite repeat markers previously shown to be informative in tumors from DNA mismatch repair–deficient mice (*Woerner et al., 2015*; *Kabbarah et al., 2003*; *Bacher et al., 2005*). Tumors were scored as MSI if 2 of 5 markers showed instability. Sodium bisulfite treatment on genomic DNA and subsequent PCR were validated and done as previously described (*Song et al., 2014*). Amplified DNA fragments were cloned using the TA Cloning Kit into the pCRII plasmid (Invitrogen, Carlsbad, CA). At least 10 clones were then randomly selected and sequenced for each sample.

## Gene expression

We tamoxifen treated 8–12 week old mice as the following: (1) Cdx2$^{fl/fl}$ mice; (2) CDX2P-CreER$^{T2}$ Apc$^{fl/fi}$ mice; (3) CDX2P-CreER$^{T2}$ Cdx2$^{fl/fl}$ mice; (4) CDX2P-CreER$^{T2}$ Braf$^{CA}$ mice; (5) CDX2P-CreER$^{T2}$ Cdx2$^{fl/fl}$ Braf$^{CA}$ mice. Three mice per group were used. Total RNA was extracted and purified from tumors or normal tissues at the colon-cecum junction. We used Affymetrix Mouse Gene 2.1 ST arrays (Affymetrix, Santa Clara, CA), which hold 41345 probe-sets, but we largely analyzed just those 25216 probe-sets that were mapped to Entrez gene IDs. Raw data was processed with the Robust

Multi-array Average algorithm (RMA). Data is log2-transformed transcript abundance estimates. We fit a one-way ANOVA model to the five groups of samples. We supply a supplementary excel workbook that holds the same data as the series matrix file, but also holds the probe-set annotation at the time we analyzed the data, and some simple statistical calculations, which selects subsets of the probe-sets as differentially expressed between pairs of groups, as well as significant $Cdx2^{-/-}$ by $BRAF^{V600E}$ interactions. It also gives the homologous human gene IDs we used for enrichment testing, which were 1-to-1 best homologs according to build 68 of NCBI's Homologene. The mouse gene expression data reported here are available from GEO (accession number GSE84650), and include the statistical analysis.

### Quantitative reverse transcription (RT)-PCR (qRT-PCR)

cDNA was synthesized using a high capacity cDNA reverse transcription kit (Applied Biosystems). qRT-PCR was performed with an ABI Prism 7300 Sequence Analyzer using a SYBR green fluorescence protocol (Applied Biosystems). Primer sequences will be provided upon request.

### Statistical analysis

One-way ANOVA model on log-transformed expression data was used to determine p values in qRT-PCR. Kaplan-Meier survival curves were compared by log-rank test. Fisher's exact test (for 2 × 2 table) and Mantel-Haenszel Chi-Square test of association (for 3 × 2 table) were used to determine significance in human CRC cases. All tests are two-sided except for one-sided Fisher Exact Tests used in *Table 2*. P-values are not corrected for multiple testing except in *Table 2*, where false discovery rates are explicitly computed. Sample sizes were chosen based on experience with previous similar experiments, except in the case of the human colon tumor samples where availability was the main constraint.

## Additional information

### Competing interests

MM: Reviewing editor, *eLife*. The other authors declare that no competing interests exist.

### Funding

| Funder | Grant reference number | Author |
| --- | --- | --- |
| National Institutes of Health | R01CA082223 | Eric R Fearon |
| National Institutes of Health | R01CA176839 | Martin McMahon |
| National Institutes of Health | P30CA046592 | Eric R Fearon |

The funders had no role in study design, data collection and interpretation, or the decision to submit the work for publication.

### Author contributions

NS, YF, Conceptualization, Data curation, Formal analysis, Investigation, Methodology, Writing—original draft, Writing—review and editing; CS, Formal analysis; YK, MG, JL, MEG, Data curation; KS, WY, Data curation, Formal analysis; MM, Resources, Writing—review and editing; KMH, JRS, NH, JKG, Resources; RK, Formal analysis, Writing—review and editing; KRC, Conceptualization; Formal analysis; Writing— original draft; Writing—review and editing; ERF, Conceptualization, Formal analysis, Supervision, Funding acquisition, Writing—original draft, Writing—review and editing

### Author ORCIDs

Jason R Spence, http://orcid.org/0000-0001-7869-3992
Eric R Fearon, http://orcid.org/0000-0003-2867-3971

### Ethics

Human subjects: The colorectal cancers were studied in accordance with the Ethical Guidelines for Human Genome/Gene Research enacted by the Japanese Government. We also studied human

benign serrated colorectal lesions obtained from the University of Michigan tissue procurement service through an Institutional Review Board-approved protocol (#00058054).

Animal experimentation: Procedures involving mice for the research described herein have been approved by the University of Michigan's Institutional Animal Care and Use Committee (PRO00005075) and were carried out according to Michigan state and US federal regulations.

## Additional files

### Supplementary files

• Supplementary file 1. Antibodies for immunohistochemistry study.

### Major datasets

The following dataset was generated:

| Author(s) | Year | Dataset title | Dataset URL | Database, license, and accessibility information |
|---|---|---|---|---|
| Sakamoto N, Feng Y, Stolfi C, Kurosu Y, Green M, Lin J, Sentani K, Yasui W, McMahon M, Hardiman KM, Spence JR, Greenson JK, Kuick R, Cho KR, Fearon ER | 2016 | Colon tumor samples from mice with Braf V600E, Cdx2-/-, or both, as well as control colon, and tumors from Apc-/- mice. | https://www.ncbi.nlm.nih.gov/geo/query/acc.cgi?acc=GSE84650 | Publicly available at the NCBI Gene Expression Omnibus (accession no: GSE84650) |

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
