## [Decision Letter]

Thank you for submitting your article "BRAFV600E cooperates with CDX2 inactivation to promote serrated colorectal tumorigenesis" for consideration by *eLife*. Your article has been reviewed by three peer reviewers, one of whom, Chi Van Dang, is a member of our Board of Reviewing Editors, and the evaluation has been overseen by Sean Morrison as the Senior Editor.

The reviewers have discussed the reviews with one another and the Reviewing Editor has drafted this decision to help you prepare a response for a possible revised submission.

Summary:

The manuscript by Sakamoto et al. describes the genomic alterations of serrated colorectal carcinoma (CRC), which has a distinct serrated pathological appearance and occurs in ~10% of CRCs. The authors studied 36 serrated CRCs and found CDX2 loss and/or BRAF mutations in 60% of serrated CRCs. Having identified CDX2 loss and BRAF mutation as hallmarks of serrated CRC, the authors modeled these losses in the mouse and were able to demonstrate cause-and-effect of these alterations and recapitulate a number of features of human serrated adenomas and adenocarcinomas in the mice (Cdx2-/-, Brafmut). The development of the model is predicated on the observation that mutational activation of Braf and loss of Cdx2 expression are common events in human serrated tumors. This particular aspect of the work does not account for a number of published studies that have described the link between Braf, Cdx2 and serrated lesions; for instance Dawson et al., 2014 (PMID: 24166180, not referenced by the authors). Similarly, comprehensive mouse modelling work from Rad et al., showed that Braf is a driver of serrated polyposis (Rad et al., 2013). The current study provides evidence that Cdx2 loss directly contributes to the progression of CRC through the serrated pathway. The model will likely be a useful tool for those studying progression of serrated colonic tumors. Unfortunately, the manuscript is descriptive and does not offer any explanation for why Cdx2 might play such a role in serrated tumor progression. Too often the authors offer correlation as a surrogate of causation, and do not provide direct evidence linking genotype to phenotype.

Essential revisions:

1) The authors are vague about the details of the animal experiments. Because the paper is driven by the development of this new model, this is important. There is no mention of tumor burden (how many tumors per mouse?), how many showed serrated features, what percentage were invasive (e.g. "Some tumor lesions in each TAM-treated… mouse showed evidence that the serrated glands had invaded the submucosa."). The paper would also be a much stronger demonstration of the new model if more than only 5 mice of the relevant genotype were produced and analysed.

2) The authors use colon organoids, a surrogate ex vivo system, but for the most part, this analysis just re-confirms gene expression changes that were already documented in the (more relevant) in vivo context. The authors are not convincing that they can recapitulate the 'serrated phenotype' in ex vivo organoid culture. In many instances, one can see the same type of cell morphology ("invaginations of epithelial cells into the organoid lumen") in normal cultures. Even if this is a true manifestation of the serrated tumor phenotype, it is a very subjective measure to rely on as the sole method of quantitation for inhibitor treatment studies (Figure 8). There is also little discussion of what this means. Is it another assay to show Braf's role in serrated tumorigenesis, or are the authors suggesting a potential specific application for Braf inhibitors in patients with serrated tumors?

3) The authors use HT29 cells to confirm CDX2/BRAF biology in human CRC, however HT29 cells carry an APC mutation, which the authors imply is a rare event in serrated tumorigenesis. More importantly, the magnitude in gene expression changes does not correlate with the animal model. For instance, ANXA10 is suppressed ~50%, in contrast to the 100-fold change seen in mice. Further, the other 3 genes analyzed show moderate (50%) changes with CDX2 overexpression alone, with only a mild increase in suppression following BRAFi treatment. Also, it is questionable that "the additive effects of CDX2 ectopic expression and BRAF or MAPK inhibition" are shown by western blot in Figure 5. There is no apparent additional suppression of PDX1 with MEK or BRAF inhibition. Finally, why were these genes chosen? Do other genes on the list show the same trend?

4) The manuscript would be greatly enriched if the authors could use informatics and their gene signature(s) to analyze the available TCGA data and compare them to the findings that loss of CDX2 expression and BRAF mutation is found in serrated CRCs that account for ~20% of cases. In essence, a retrospective bioinformatics study of as many as possible CRC TCGA data to identify the serrated subset of CRCs purely based on informatics would be highly instructive.

---

## [Author Response]

*Essential revisions:*

*1) The authors are vague about the details of the animal experiments. Because the paper is driven by the development of this new model, this is important. There is no mention of tumor burden (how many tumors per mouse?), how many showed serrated features, what percentage were invasive (e.g. "Some tumor lesions in each TAM-treated… mouse showed evidence that the serrated glands had invaded the submucosa."). The paper would also be a much stronger demonstration of the new model if more than only 5 mice of the relevant genotype were produced and analysed.*

We have modified the manuscript by the inclusion of a new supplementary table ([Supplementary-material SD3-data]) that provides data on the number and sizes of the tumor masses arising in seven independent mice where *Cdx2* was inactivated together with BRAF^V600E^ activation in epithelium of the terminal ileum, cecum, and colon. The invasive features in the lesions are also clearly defined in the table in [Supplementary-material SD3-data] and the text. The survival differences we presented in the original manuscript were indeed highly statistically significant. However, we recognize the reviewer team’s points about including additional mice. We now include data on seven CDX2-CreER^T2^ Cdx2^fl/fl^ Braf^CA^ mice and six CDX2P-CreER^T2^ Cdx2^fl/fl^ mice, along with five CDX2P-CreER^T2^ Braf^CA^ mice and five CDX2P-CreER^T2^ Apc^fl/+^Cdx2^fl/fl^ Braf^CA^ mice in Figure 2. The statistical significance of the data (P<0.002) is clear, demonstrating that combined defects of *Cdx2* and *Braf* lead to dramatically shorter survival than *Cdx2* or *Braf* defects individually.

*2) The authors use colon organoids, a surrogate* ex vivo *system, but for the most part, this analysis just re-confirms gene expression changes that were already documented in the (more relevant)* in vivo *context. The authors are not convincing that they can recapitulate the 'serrated phenotype' in* ex vivo *organoid culture. In many instances, one can see the same type of cell morphology ("invaginations of epithelial cells into the organoid lumen") in normal cultures. Even if this is a true manifestation of the serrated tumor phenotype, it is a very subjective measure to rely on as the sole method of quantitation for inhibitor treatment studies (Figure 8). There is also little discussion of what this means. Is it another assay to show Braf's role in serrated tumorigenesis, or are the authors suggesting a potential specific application for Braf inhibitors in patients with serrated tumors?*

The use of colon organoid models was presented to indicate that selected morphological alterations – serrated glandular features and particular nuclear alterations – seen in CDX2^null^/BRAF^V600E^epithelium in mice and that were akin to signature morphological features in human serrated morphology colorectal cancer (CRCs) could also be seen in the organoid model system. The ex vivo organoid model system allows for studies of selected serrated CRC phenotypic features, such as analysis of how well the gene expression patterns seen in vivo in human and mouse serrated colon tumors can be recapitulated in an ex vivo model system. We recognize the organoid system and morphology endpoints, as presented, only moderately extended on the in vivo analyses. In the revised manuscript, we pursued additional studies and provide new data addressing the issue of how well the organoid system for CDX2^null^/BRAF^V600E^epithelium can recapitulate gene expression patterns seen in mouse CDX2^null^/BRAF^V600E^tumor tissues or human serrated tumors. As we now present in the manuscript text and a new Figure 8, only a subset of the genes dramatically upregulated in mouse CDX2^null^/BRAF^V600E^colon epithelium and human serrated colon cancers are similarly dramatically upregulated in the CDX2^null^/BRAF^V600E^organoid model system. Our findings suggest that the ex vivo organoid model system cannot fully recapitulate the gene expression patterns seen in serrated mouse and human colorectal tumors, further highlighting the importance of in vivo models to study how distinct gene lesions can functionally cooperate to generate tumor phenotypes. We present data in the revised Figure 8 to show that pharmacologic inhibition of MEK signaling can dramatically inhibit the expression of four genes that were potently up-regulated by the cooperative interactions between *Cdx2* inactivation and *Braf^V600E^*activation. The intent of the MEK pharmacological inhibition studies was simply to show that downstream inhibition of signaling strongly inhibits the ability of Braf^V600E^ to cooperate with Cdx2 inactivation in regulation of gene expression. Future studies may address how BRAF inhibitors might be used for patients whose serrated tumors harbor BRAFV600E defects, but we do not emphasize potential therapeutic issues related to BRAF inhibition in our manuscript.

*3) The authors use HT29 cells to confirm CDX2/BRAF biology in human CRC, however HT29 cells carry an APC mutation, which the authors imply is a rare event in serrated tumorigenesis. More importantly, the magnitude in gene expression changes does not correlate with the animal model. For instance, ANXA10 is suppressed ~50%, in contrast to the 100-fold change seen in mice. Further, the other 3 genes analyzed show moderate (50%) changes with CDX2 overexpression alone, with only a mild increase in suppression following BRAFi treatment. Also, it is questionable that "the additive effects of CDX2 ectopic expression and BRAF or MAPK inhibition" are shown by western blot in Figure 5. There is no apparent additional suppression of PDX1 with MEK or BRAF inhibition. Finally, why were these genes chosen? Do other genes on the list show the same trend?*

We recognize the reviewer team’s concerns about the limitations of the HT29 cell line as a model of serrated tumorigenessis and the modest effects of CDX2 restoration and BRAFV600E inhibition on the expression of ANXA10 and the other genes studied, relative to the gene expression patterns seen in CDX2^null^/BRAF^V600E^-mutant epithelium in the mouse. Beyond the issue of the APC mutation in HT29 cells, a challenge with the approach of using the HT29 human colon cancer cell line to study cooperation of CDX2 and BRAF^V600E^ in modulating gene expression is that the approaches used in vitro are the inverse of the situation in in vivo; i.e., combined Cdx2 inactivation and Braf^V600E^ activation in vivo versus ectopic CDX2 expression and/or BRAF inhibition in the HT29 in vitro system. As noted above in our response to point #2, we have now used CDX2^null^/BRAF^V600E^ organoids to undertake studies to define the effects of MEK inhibitors downstream of BRAF activation on selected genes that were potently activated in CDX2^null^/BRAF^V600E^ mouse colon epithelium (See [Supplementary-material SD7-data] for list of genes that were chosen), and these new data are presented in an entirely new Figure 8, as discussed above in Point #2. Because of the reviewer concerns about the limitation of the HT29 studies and data, we have removed them from the revised manuscript.

*4) The manuscript would be greatly enriched if the authors could use informatics and their gene signature(s) to analyze the available TCGA data and compare them to the findings that loss of CDX2 expression and BRAF mutation is found in serrated CRCs that account for ~20% of cases. In essence, a retrospective bioinformatics study of as many as possible CRC TCGA data to identify the serrated subset of CRCs purely based on informatics would be highly instructive.*

To demonstrate cooperative effects of CDX2 and BRAF function in regulating gene expression in human CRC, we compared our mouse gene signature to 212 human CRC samples available in TCGA project data (https://gdac.broadinstitute.org/) that also have mutation data. Among these samples, we compared the 18 samples that had low CDX2 and V600E mutations to the 104 samples with high CDX2 that were not BRAF mutant, and selected the genes with p-values of <0.01 (two-sample T-test) and fold-changes of >1.3 between the two groups. Then, we computed the intersection of this selection to the similar selection we had performed in our mouse data to ask for significant Cdx2 by Braf interactions. We observed strong association between our mouse data and human CRC data with significant enrichment of genes found “up” in both data-sets as well as “down” in both data-sets, and few disagreements between both data-sets (p=4.5x10^-49^, Mantel-Haenszel Chi-Square test of association). We now present these new data in a significantly revised figure and new supplement (i.e., Figure 5 and [Supplementary-material SD6-data]). In addition, we also compared our mouse gene signature to expression data for 8 serrated and 29 conventional colorectal cancers obtained from GEO series GSE4045 (called Finnish data) (Laiho et al. 2007). Again, the genes that showed strong cooperative interaction between Cdx2 loss and Braf^V600E^ expression in the mouse colon tumors (either up or down) were found to be significantly enriched in the human serrated CRCs vs. conventional CRCs (p=1.4x10^-11^) (presented in a new [Supplementary-material SD6-data]). Our findings suggest that the gene signature found in our Cdx2^-/-^Braf^V600E^ mouse model is highly instructive to identify the serrated subset of human CRCs, either in the TCGA data or in other available data sets.